# BanditPAM++: Faster $k$-medoids Clustering

**Mo Tiwari**
Stanford University
motiwari@stanford.edu

**Ryan Kang***
Stanford University
txryank@stanford.edu

**Donghyun Lee***
University College London
donghyun.lee.21@ucl.ac.uk

**Sebastian Thrun**
Stanford University
thrun@stanford.edu

**Chris Piech**
Stanford University
piech@cs.stanford.edu

**Ilan Shomorony‡**
University of Illinois at Urbana-Champaign
ilans@illinois.edu

**Martin Jinye Zhang‡**
Carnegie Mellon University
martinzh@andrew.cmu.edu

## Abstract

Clustering is a fundamental task in data science with wide-ranging applications. In $k$-medoids clustering, cluster centers must be actual datapoints and arbitrary distance metrics may be used; these features allow for greater interpretability of the cluster centers and the clustering of exotic objects in $k$-medoids clustering, respectively. $k$-medoids clustering has recently grown in popularity due to the discovery of more efficient $k$-medoids algorithms. In particular, recent research has proposed BanditPAM, a randomized $k$-medoids algorithm with state-of-the-art complexity and clustering accuracy. In this paper, we present BanditPAM++, which accelerates BanditPAM via two algorithmic improvements, and is $O(k)$ faster than BanditPAM in complexity and substantially faster than BanditPAM in wall-clock runtime. First, we demonstrate that BanditPAM has a special structure that allows the reuse of clustering information *within* each iteration. Second, we demonstrate that BanditPAM has additional structure that permits the reuse of information *across* different iterations. These observations inspire our proposed algorithm, BanditPAM++, which returns the same clustering solutions as BanditPAM but often several times faster. For example, on the CIFAR10 dataset, BanditPAM++ returns the same results as BanditPAM but runs over $10\times$ faster. Finally, we provide a high-performance C++ implementation of BanditPAM++, callable from Python and R, that may be of interest to practitioners at https://github.com/motiwari/BanditPAM. Auxiliary code to reproduce all of our experiments via a one-line script is available at https://github.com/ThrunGroup/BanditPAM_plusplus_experiments/.

## 1 Introduction

Clustering is a critical and ubiquitous task in data science and machine learning. Clustering aims to separate a dataset $\mathcal{X}$ of $n$ datapoints into $k$ disjoint sets that form a partition of the original dataset. Intuitively, datapoints within a cluster are similar and datapoints in different clusters are dissimilar. Clustering problems and algorithms have found numerous applications in textual data [20], social network analysis [18], biology [28], and education [28].

A common objective function used in clustering is Equation 1:

$$L(\mathcal{C}) = \sum_{j=1}^{n} \min_{c \in \mathcal{C}} d(c, x_j). \tag{1}$$

Under this loss function, the goal becomes to minimize the distance, defined by the distance function $d$, from each datapoint to its nearest cluster center $c$ among the set of cluster centers $\mathcal{C}$. Note that this formulation is general and does not require the datapoints to be vectors or assume a specific functional form of the distance function $d$.

Specific choices of $\mathcal{C}$, dataset $\mathcal{X}$, and distance function $d$ give rise to different clustering problems. Perhaps one of the most commonly used clustering methods is $k$-means clustering [17, 16]. In $k$-means clustering, each datapoint is a vector in $\mathbb{R}^p$ and the distance function $d$ is usually taken to be squared $L_2$ distance; there are no constraints on $\mathcal{C}$ other than that it is a subset of $\mathbb{R}^p$. Mathematically, the $k$-means objective function is

$$L(\mathcal{C}) = \sum_{j=1}^{n} \min_{c \in \mathcal{C}} \|x_j - c\|_2^2 \tag{2}$$

subject to $|\mathcal{C}| = k$.

The most common algorithm for the $k$-means problem is Lloyd iteration [16], which has been improved by other algorithms such as $k$-means++ [2]. These algorithms are widely used in practice due to their simplicity and computational efficiency. Despite widespread use in practice, $k$-means clustering suffers from several limitations. Perhaps its most significant restriction is the choice of $d$ as the squared $L_2$ distance. This choice of $d$ is for computational efficiency – as the mean of many points can be efficiently computed under squared $L_2$ distance – but prevents clustering with other distance metrics that may be preferred in other contexts [22, 8, 5]. For example, $k$-means is difficult to use on textual data that necessitates string edit distance [20], social network analyses using graph metrics [18], or sparse datasets (such as those found in recommendation systems [15]) that lend themselves to other distance functions. While $k$-means algorithms have been adapted to specific other metrics, e.g., cosine distance [23], these methods are bespoke to the metric and not readily generalizable. Another limitation of $k$-means clustering is that the set of cluster centers $\mathcal{C}$ may often be uninterpretable, as each cluster center is (generally) a linear combination of datapoints. This limitation can be especially problematic when dealing with structured data, such as parse trees in context-free grammars, where the mean of trees is not necessarily well-defined, or images in computer vision, where the mean image can appear as random noise [28, 15].

In contrast with $k$-means clustering, $k$-*medoids* clustering [10, 11] requires the cluster centers to be actual datapoints, i.e., requires $\mathcal{C} \subset \mathcal{X}$. More formally, the objective is to find a set of *medoids* $\mathcal{M} \subset \mathcal{X}$ (versus $\mathbb{R}^p$ in $k$-means) that minimizes

$$L(\mathcal{M}) = \sum_{j=1}^{n} \min_{m \in \mathcal{M}} d(m, x_j) \tag{3}$$

subject to $|\mathcal{M}| = k$. Note that there is no restriction on the distance function $d$.

$k$-medoids clustering has several advantages over $k$-means. Crucially, the requirement that each cluster center is a datapoint leads to greater interpretability of the cluster centers because each cluster center can be inspected. Furthermore, $k$-medoids supports arbitrary dissimilarity measures; the distance function $d$ in Equation (3) need not be a proper metric, i.e., may be negative, asymmetric, or violate the triangle inequality. Because $k$-medoids supports arbitrary dissimilarity measures, it can also be used to cluster "exotic" objects that are not vectors in $\mathbb{R}^p$, such as trees and strings [28], without embedding them in $\mathbb{R}^p$.

The $k$-medoids clustering problem in Equation (3) is a combinatorial optimization algorithm that is NP-hard in general [25]; as such, algorithms for $k$-medoids clustering are restricted to heuristic solutions. A popular early heuristic solution for the $k$-medoids problem was the Partitioning Around Medoids (PAM) algorithm [11]; however, PAM is quadratic in dataset size $n$ in each iteration, which

is prohibitively expensive on large dataset. Improving the computational efficiency of these heuristic solutions is an active area of research. Recently, [28] proposed BanditPAM, the first subquadratic algorithm for the $k$-medoids problem that matched prior state-of-the-art solutions in clustering quality. In this work, we propose BanditPAM++, which improves the computational efficiency of BanditPAM while maintaining the same results. We anticipate these computational improvements will be important in the era of big data, when $k$-medoids clustering is used on huge datasets.

**Contributions:** We propose a new algorithm, BanditPAM++, that is significantly more computationally efficient than PAM and BanditPAM but returns the same clustering results with high probability. BanditPAM++ is $O(k)$ faster than BanditPAM in complexity and substantially faster than BanditPAM in actual runtime wall-clock runtime. Consequently, BanditPAM++ is faster than prior state-of-the-art $k$-medoids algorithms while maintaining the same clustering quality. BanditPAM++ is based on two observations about the structure of BanditPAM and the $k$-medoids problem, described in Section 4. The first observation leads to a technique that we call *Virtual Arms (VA)*. The second observation leads to a technique that we refer to as *Permutation-Invariant Caching (PIC)*. We combine these techniques in BanditPAM++ and prove (in Section 5) and experimentally validate (in Section 6) that BanditPAM++ returns the same solution to the $k$-medoids problem as PAM and BanditPAM with high probability, but is more computationally efficient. In some instances, BanditPAM++ is over $10\times$ faster than BanditPAM. Additionally, we provide a highly optimized implementation of BanditPAM++ in C++ that is callable from Python and R and may be of interest to practitioners.

## 2   Related Work

As discussed in Section 1, global optimization of the $k$-medoids problem (Equation (3)) is NP-hard in general [25]. Recent work attempts to perform attempts global optimization and is able to achieve an optimality gap of 0.1% on one million datapoints, but is restricted to $L_2$ distance and takes several hours to run on commodity hardware [24].

Because of the difficulty of global optimization of the $k$-medoids problem, many heuristic algorithms have been developed for the $k$-medoids problem that scale polynomially with the dataset size and number of clusters. The complexity of these algorithms is measured by their sample complexity, i.e., the number of pairwise distance computations that are computed; these computations have been observed to dominate runtime costs and, as such, sample complexity translates to wall-clock time via an approximately constant factor [28] (this is also consistent with our experiments in Section 6 and Appendix 3).

Among the heuristic solutions for the $k$-medoids problem, the algorithm with the best clustering loss is Partitioning Around Medoids (PAM) [10, 11], which consists of two phases: the BUILD phase and the SWAP phase. However, the BUILD phase and each SWAP iteration of PAM perform $O(kn^2)$ distance computations, which can be impractical for large datasets or when distance computations are expensive. We provide greater details about the PAM algorithm in Section 3 because it is an important baseline against which we assess the clustering quality of new algorithms.

Though PAM achieves the best clustering loss among heuristic algorithms, the era of huge data has necessitated the development of faster $k$-medoids algorithms in recent years. These algorithms have typically been divided into two categories: those that agree with PAM and recover the same solution to the $k$-medoids problem but scale quadratically in $n$, and those that sacrifice clustering quality for runtime improvements. In the former category, [25] proposed a deterministic algorithm called FastPAM1, which maintains the same output as PAM but reduces the computational complexity of each SWAP iteration from $O(kn^2)$ to $O(n^2)$. However, this algorithm still scales quadratically in $n$ in every iteration, which is prohibitively expensive on large datasets.

Faster heuristic algorithms have been proposed but these usually sacrifice clustering quality; such algorithms include CLARA [11], CLARANS [21], and FastPAM [25]. While these algorithms scale subquadratically in $n$, they return substantially worse solutions than PAM [28]. Other algorithms with better sample complexity, such as optimizations for Euclidean space and those based on tabu search heuristics [6] also return worse solutions. Finally, [1] attempts to minimize the number of *unique* pairwise distances or adaptively estimate these distances or coordinate-wise distances in specific settings [14, 3], but all these approaches sacrifice clustering quality for runtime.

Recently, [28] proposed BanditPAM, a state-of-the-art $k$-medoids algorithm that arrives at the same solution as PAM with high probability in $O(kn \log n)$ time. BanditPAM borrows techniques from the multi-armed bandit literature to sample pairwise distance computations rather than compute all $O(n^2)$. In this work, we show that BanditPAM can be made more efficient by reusing distance computations both *within* iterations and *across* iterations.

We note that the use of adaptive sampling techniques and multi-armed bandits to accelerate algorithms has also had recent successes in other work, e.g., to accelerate the training of Random Forests [27], solve the Maximum Inner Product Search problem [27], and more [26].

## 3 Preliminaries and Background

**Notation:** We consider a dataset $\mathcal{X}$ of size $n$ (that may contain vectors in $\mathbb{R}^p$ or other objects). Our goal is to find a solution to the $k$-medoids problem, Equation (3). We are also given a dissimilarity function $d$ that measures the dissimilarity between two objects in $\mathcal{X}$. Note that we do not assume a specific functional form of $d$. We use $[n]$ to denote the set $\{1, \ldots, n\}$, and $a \wedge b$ (respectively, $a \vee b$) to denote the minimum (respectively, maximum) of $a$ and $b$.

**Partitioning Around Medoids (PAM):** The original Partitioning Around Medoids (PAM) algorithm [10, 11] consists of two main phases: BUILD and SWAP. In the BUILD phase, PAM iteratively initializes each medoid in a greedy, one-by-one fashion: in each iteration, it selects the next medoid that would reduce the $k$-medoids clustering loss (Equation (3)) the most, given the prior choices of medoids. More precisely, given the current set of $l$ medoids $\mathcal{M}_l = \{m_1, \cdots, m_l\}$, the next point to add as a medoid is:

$$m^* = \underset{x \in \mathcal{X} \setminus \mathcal{M}_l}{\arg\min} \sum_{j=1}^{n} \left[ d(x, x_j) \wedge \min_{m' \in \mathcal{M}_l} d(m', x_j) \right] \tag{4}$$

The output of the BUILD step is an initial set of the $k$ medoids, around which a local search is performed by the SWAP phase. The SWAP phase involves iteratively examining all $k(n-k)$ medoid-nonmedoid pairs and performs the swap that would lower the total loss the most. More precisely, with $\mathcal{M}$ the current set of $k$ medoids, PAM finds the best medoid-nonmedoid pair to swap:

$$(m^*, x^*) = \underset{(m,x) \in \mathcal{M} \times (\mathcal{X} \setminus \mathcal{M})}{\arg\min} \sum_{j=1}^{n} \left[ d(x, x_j) \wedge \min_{m' \in \mathcal{M} \setminus \{m\}} d(m', x_j) \right] \tag{5}$$

PAM requires $O(kn^2)$ distance computations for the $k$ greedy searches in the BUILD step and $O(kn^2)$ distance computations for each SWAP iteration [28]. The quadratic complexity of PAM makes it prohibitively expensive to run on large datasets. Nonetheless, we describe the PAM algorithm because it has been observed to have the best clustering loss among heuristic solutions to the $k$-medoids problem. More recent algorithms, such as BanditPAM [28], achieve the same clustering loss but have a significantly improved complexity of $O(kn \log n)$ in each step. Our proposed algorithm, BanditPAM++, improves upon the computational complexity of BanditPAM by a factor of $O(k)$.

**Sequential Multi-Armed Bandits:** BanditPAM [28] improves the computational complexity of PAM by converting each step of PAM to a multi-armed bandit problem. A multi-armed bandit problem (MAB) is defined as a collection of random variables $\{R_1, \ldots, R_n\}$, called actions or arms. We are commonly interested in the best-arm identification problem, which is to identify the arm with the highest mean, i.e., $\arg\max_i \mathbb{E}[R_i]$, with a given probability of possible error $\delta$. Many algorithms for this problem exist, each of which make distributional assumptions about the random variables $\{R_1, \ldots, R_n\}$; popular ones include the upper confidence bound (UCB) algorithm and successive elimination. For an overview of common algorithms, we refer the reader to [9].

We define a *sequential multi-armed bandit problem* to be an ordered sequence of multi-armed bandit problems $Q = \{B_1, \ldots, B_T\}$ where each individual multi-armed bandit problem $B_t = \{R_1^t, \ldots, R_n^t\}$ has the same number of arms $n$, with respective, timestep-dependent means $\mu_1^t, \ldots, \mu_n^t$. At each timestep $t$, our goal is to determine (and take) the best action $a_t = \arg\max_i \mathbb{E}[R_i^t]$. Crucially, our choices of $a_t$ will affect the rewards at future timesteps, i.e., the $R_i^{t'}$ for $t' > t$. Our definition of a sequential multi-armed bandit problem is similar to non-stationary multi-armed bandit problems, with the added restriction that the only non-stationarity in the problem comes from our previous actions.

We now make a few assumptions for tractability. We assume that each $R_i^t$ is observed by sampling an element from a set $\mathcal{S}$ with $S$ possible values. We refer to the values of $\mathcal{S}$ as the *reference points*, where each possible reference point is sampled with equal probability and determines the observed reward. With some abuse of notation, we write $R_i^t(x_s)$ for the reward observed from arm $R_i^t$ when the latent variable from $\mathcal{S}$ is observed to be $x_s$. We refer to a sequential multi-armed bandit as *permutation-invariant*, or as a *SPIMAB* (for S̲equential, P̲ermutation-I̲nvariant M̲ulti-A̲rmed B̲andit), if the following conditions hold:

1. For every arm $i$ and timestep $t$, $R_i^t = f(D_i, \{a_0, a_1, \ldots, a_{t-1}\})$ for some known function $f$ and some random variable $D_i$ with mean $\mu_i := \mathbb{E}[D_i] = \frac{1}{S} \sum_{s=1}^{S} D_i(x_s)$,

2. There exists a common set of reference points, $\mathcal{S}$, shared amongst each $D_i$,

3. It is possible to sample each $D_i$ in $O(1)$ time by drawing from the points in $\mathcal{S}$ without replacement, and

4. $f$ is computable in $O(1)$ time given its inputs.

Intuitively, the conditions above require that at each timestep, each random variable $R_i^t$ is expressible as a known function of another random variable $D_i$ and the prior actions taken in the sequential multi-armed bandit problem. Crucially, $D_i$ *does not depend on the timestep*; $R_i^t$ is only permitted to depend on the timestep through the agent's previously taken actions $\{a_0, a_1, \ldots, a_{t-1}\}$. The SPIMAB conditions imply that if $\mathbb{E}[D_i] = \mu_i$ is known for each $i$, then $\mu_i^t := \mathbb{E}[R_i^t]$ is also computable in $O(1)$ time for each $i$ and $t$, i.e., for each arm and timestep.

**BanditPAM:** BanditPAM [28] reduces the scaling with $n$ of each step of the PAM algorithm by reformulating each step as a best-arm identification problem. In PAM, each of the $k$ BUILD steps has complexity $O(n^2)$ and each SWAP iteration has complexity $O(kn^2)$. In contrast, the complexity of BanditPAM is $O(n \log n)$ for each of the $k$ BUILD steps and $O(kn \log n)$ for each SWAP iteration. Fundamentally, BanditPAM achieves this reduction in complexity by sampling distance computations instead of using all $O(n^2)$ pairwise distances in each iteration. We note that all $k$ BUILD steps of BanditPAM (respectively, PAM) have the same complexity as *each* SWAP iteration of BanditPAM (respectively, PAM). Since the number of SWAP iterations is usually $O(k)$ ([28]; see also our experiments in the Appendix 3), most of BanditPAM's runtime is spent in the SWAP iterations; this suggests improvements to BanditPAM should focus on expediting its SWAP phase.

## 4   BanditPAM++: Algorithmic Improvements to BanditPAM

In this section, we discuss two improvements to the BanditPAM algorithm. We first show how each SWAP iteration of BanditPAM can be improved via a technique we call Virtual Arms (VA). With this improvement, the modified algorithm can be cast as a SPIMAB. The conversion to a SPIMAB permits a second improvement via a technique we call the Permutation-Invariant Cache (PIC). Whereas the VA technique improves only the SWAP phase, the PIC technique improves both the BUILD and SWAP phases. The VA technique improves the complexity of each SWAP iteration by a factor of $O(k)$, whereas the PIC improves the wall-clock runtime of both the BUILD and SWAP phases.

### 4.1   Virtual Arms (VA)

As discussed in Section 3, most of the runtime of BanditPAM is spent in the SWAP iterations. When evaluating a medoid-nonmedoid pair $(m, x_i)$ to potentially swap, BanditPAM estimates the quantity:

$$\Delta L_{m,x_i} = \frac{1}{n} \sum_{s=1}^{n} \Delta l_{m,x_i}(x_s), \tag{6}$$

for each medoid $m$ and candidate nonmedoid $x_i$, where

$$\Delta l_{m,x_i}(x_s) = \left( d(x_i, x_s) - \min_{m' \in \mathcal{M} \setminus \{m\}} d(m', x_s) \right) \wedge 0 \tag{7}$$

is the change in clustering loss (Equation (3)) induced on point $x_s$ for swapping medoid $m$ with nonmedoid $x_i$ in the set of medoids $\mathcal{M}$. Crucially, we will find that for a given $x_s$, each $\Delta l_{m,x_i}(x_s)$ for $m = 1, \ldots, k$, except possibly one, is equal. We state this observation formally in Theorem 1.

| General SPIMAB Term | BanditPAM++ BUILD step | BanditPAM++ SWAP step |
|---|---|---|
| Arms, $\{R_i^t\}_{j=1}^n$ | Candidate points for medoids | Points to swap in as medoids |
| Reference points, $\mathcal{S}$ | Points of dataset $\mathcal{X}$ | Points of dataset $\mathcal{X}$ |
| Timestep, $t$ | $t$-th medoid to be added | $(t-k)$-th swap to be performed |
| $D_i(x_s)$ | Distance between $x_i$ and $x_s$ | Distance between $x_i$ and $x_s$ |
| $f(D_i, \{a_0, a_1, \ldots, a_{t-1}\})$ | Equation 8 | Equation 8 |

Table 1: BanditPAM++'s two phases can each be cast in the SPIMAB framework.

**Theorem 1.** *Let $\Delta l_{m,x_i}(x_s)$ be the change in clustering loss induced on point $x_s$ by swapping medoid $m$ with nonmedoid $x_i$, given in Equation (7), with $x_s$ and $x_i$ fixed. Then the values $\Delta l_{m,x_i}(x_s)$ for $m = 1, \ldots, k$ are equal, except possibly where $m$ is the medoid for reference point $x_s$.*

Theorem 1 is proven in Appendix 2. Crucially, Theorem 1 tells us that when estimating $\Delta L_{m,x_i}$ in Equation (6) for fixed $x_i$ and various values of $m$, we can reuse a significant number of the summands across different indices $m$ (across $k$ medoids). We note that Theorem 1 has been observed in alternate forms, e.g., as the "FastPAM1" trick, in prior work [28]. However, to the best of our knowledge, we are the first to provide a formal statement and proof of Theorem 1 and demonstrate its use in an adaptive sampling scheme inspired by multi-armed bandits.

Motivated by Theorem 1, we present the SWAP step of our algorithm, BanditPAM++, in Algorithm 1. BanditPAM++ uses the VA technique to improve the complexity of each SWAP iteration by a factor of $O(k)$. We call the technique "virtual arms" because it uses only a constant number of distance computations to update each of the $k$ "virtual" arms for each of the "real" arms, where a "real" arm corresponds to a datapoint.

## 4.2 Permutation-Invariant Caching (PIC):

The original BanditPAM algorithms considers each of the $k(n-k)$ medoid-nonmedoid pairs as arms. With the VA technique described in Section 4.1, BanditPAM++ instead considers each of the $n$ datapoints (including existing medoids) as arms in each SWAP iteration. Crucially, this implies that the BUILD phase and each SWAP iteration of BanditPAM++ consider the same set of arms. It is this observation, induced by the VA technique, that allows us to cast BanditPAM++ as a SPIMAB and implement a second improvement. We call this second improvement the Permutation-Invariant Cache (PIC).

We formalize the reduction of BanditPAM++ to a SPIMAB in Table 1. In the SPIMAB formulation of BanditPAM++, the set of reference points $\mathcal{S}$ is the same as the set of datapoints $\mathcal{X}$. Each $D_i$ is a random variable representing the distance from point $x_i$ to one of the sampled reference points and can be sampled in $O(1)$ time without replacement. Each $\mu_i = \mathbb{E}[D_i]$ corresponds to the average distance from point $x_i$ to all the points in the dataset $\mathcal{X}$. Each arm $R_i^t$ corresponds to the point that we would add to the set of medoids (for $t \leq k$) or swap in to the set of medoids (for $t > k$), of which there are $n$ at each possible timestep. Similarly, the actions $\{a_0, \ldots, a_p, \ldots, a_t\}$ correspond to points added to the set of medoids (for $t \leq k$) or swaps performed (for $t > k$). Equation (8) provides the functional forms of $f$ for each of the BUILD and SWAP steps.

$$f(D_i(x_s), \mathcal{A}) = \left( d(x_i, x_s) - \min_{m' \in \mathcal{M}} d(m', x_s) \right) \wedge 0. \tag{8}$$

where $\mathcal{M}$ is a set of medoids. For the BUILD step, $\mathcal{A}$ is a sequence of $t$ actions that results in a set of medoids of size $t \leq k$, and, for the SWAP step, $\mathcal{A}$ is a set of actions that results in a set of medoids $\mathcal{M}$ of size $k$.

The observation that BanditPAM++ is a SPIMAB allows us to develop an intelligent cache design, which we call a *permutation-invariant cache (PIC)*. We may choose a permutation $\pi$ of the reference points $\mathcal{S} = \mathcal{X}$ and sample distance computations to these reference points in the order of the permutation. Since we only need to sample some of the reference points, and not all of them, we do not need to compute all $O(n^2)$ pairwise distances. Crucially, we can also reuse distance computations across different steps of BanditPAM++ to save on computational cost and runtime.

---

**Algorithm 1** BanditPAM++ SWAP Step ( $f_j(D_j, \{a_1, \ldots, a_t\})$, $\delta$, $\sigma_x$, permutation $\pi$ of $[n]$ )

---

1: $\mathcal{S}_{\text{solution}} \leftarrow [n]$        ▷ Set of potential solutions to MAB
2: $t' \leftarrow 0$        ▷ Number of reference points evaluated
3: For all $(i, j) \in [n] \times [k]$, set $\hat{\mu}_{i,j} \leftarrow 0$, $C_{i,j} \leftarrow \infty$      ▷ Initial means and CIs for all swaps
4: **while** $t' < n$ and $|\mathcal{S}_{\text{solution}}| > 1$ **do**
5:      $s \leftarrow \pi(t')$        ▷ Uses PIC
6:      **for all** $i \in \mathcal{S}_{\text{solution}}$ **do**
7:          Let $c(s)$ and $c^{(2)}(s)$ be the indices of $x_s$'s closest and second closest medoids   ▷ Cached
8:          Compute distance to $x_s$'s closest medoid $d_1 := d(m_{c(s)}, x_s)$      ▷ Cached
9:          Compute distance to $x_s$'s second closest medoid $d_2 := d(m_{c^{(2)}(s)}, x_s)$     ▷ Cached
10:         Compute $d_i := d(x_i, x_s)$      ▷ Reusing $x_s$'s across calls leads to more cache hits
11:         $\hat{\mu}_{i,c(s)} \leftarrow \frac{t'\hat{\mu}_{i,c(s)} - d_1 + \min(d_2, d_i)}{t'+1}$        ▷ Update running mean for $x_s$'s medoid
12:         $C_{i,c(s)} \leftarrow \sigma_i \sqrt{\frac{\log(\frac{1}{\delta})}{t'+1}}$       ▷ Update confidence interval for $x_s$'s medoid
13:         **for all** $j \in \{1, \ldots, k\} \setminus \{c(s)\}$ **do**
14:           $\hat{\mu}_{i,j} \leftarrow \frac{t'\hat{\mu}_{i,j} + f(D_i(x_s), a_1, \ldots, a_k)}{t'+1}$     ▷ Update running means; does not depend on $j$
15:           $C_{i,j} \leftarrow \sigma_i \sqrt{\frac{\log(\frac{1}{\delta})}{t'+1}}$       ▷ Update confidence intervals; does not depend on $j$
16:     $\mathcal{S}_{\text{solution}} \leftarrow \{i : \exists j \text{ s.t. } \hat{\mu}_{i,j} - C_{i,j} \leq \min_{i,j}(\hat{\mu}_{i,j} + C_{i,j})\}$     ▷ Filter suboptimal arms
17:     $t' \leftarrow t' + 1$
18: **if** $|\mathcal{S}_{\text{solution}}| = 1$ **then**
19:     **return** $i^* \in \mathcal{S}_{\text{solution}}$ and $j^* = \arg\min_j \mu_{\hat{i^*}, j}$
20: **else**
21:     Compute $\mu_{i,j}$ exactly for all $i \in \mathcal{S}_{\text{solution}}$      ▷ At most $3n$ distance computations
22:     **return** $(i^*, j^*) = \arg\min_{(i,j): i \in \mathcal{S}_{\text{solution}}} \mu_{i,j}$

---

The full BanditPAM++ algorithm is given in Algorithm 1. Crucially, for each candidate point $x_i$ to swap into the set of medoids on Line 6, we only perform 3 distance computations (not $k$) to update all $k$ arms, each of which has a mean and confidence interval (CI), on Lines 13-15. This is permitted by Theorem 1 and the VA technique which says that $k - 1$ "virtual" arms for a fixed $i$ will get the same update. The PIC technique allows us to choose a permutation of reference points (the $x_s$'s) and reuse those $x_s$'s across the BUILD and SWAP steps; as such, many values of $d(x_i, x_s)$ can be cached.

We emphasize that BanditPAM++ uses the same BUILD step as the original BanditPAM algorithm, but with the PIC. The PIC is also used during the SWAP step of BanditPAM++, as is the VA technique. We prove that the full BanditPAM++ algorithm returns the same results as BanditPAM and PAM in Section 5 and demonstrate the empirical benefits of both the PIC and VA techniques in Section 6.

## 5 Analysis of the Algorithm

In this section, we demonstrate that, with high probability, BanditPAM++ returns the same answer to the $k$-medoids clustering problem as PAM and BanditPAM while improving the SWAP complexity of BanditPAM by $O(k)$ and substantially decreasing its runtime. Since the BUILD step of BanditPAM is the same as the BUILD step of BanditPAM++, it is sufficient to show that each SWAP step of BanditPAM++ returns the same swap as the corresponding step of BanditPAM (and PAM). All of the following theorems are proven in Appendix 2.

First, we demonstrate that PIC does not affect the results of BanditPAM++ in Theorem 2:

**Theorem 2.** *Let* $\mathcal{X} = \{x_1, \ldots, x_S\}$ *be the reference points of* $D_i$, *and let* $\pi$ *be a random permutation of* $\{1, \ldots, S\}$. *Then for any* $c \leq S$, $\sum_{q=1}^{c} D_i(x_{\pi(p_q)})$ *has the same distribution as* $\sum_{q=1}^{c} D_i(x_{p_q})$, *where each* $p_q$ *is drawn uniformly without replacement from* $\{1, \ldots, S\}$.

Intuitively, Theorem 2 says that instead of randomly sampling new reference points at each iteration of BanditPAM++, we may choose a fixed permutation $\pi$ in advance and sample in permutation order at each step of the algorithm. This allows us to reuse computation across different steps of the algorithm.

We now show that BanditPAM++ returns the same result as BanditPAM (and PAM) in every SWAP iteration and has the same complexity in $n$ as BanditPAM. First, we consider a single call to Algorithm 1. Let $\mu_i := \min_{j \in [k]} \mu_{i,j}$ and let $i^* := \arg\min_{i \in [n]} \mu_i$ be the optimal point to swap in to the set of medoids, so that the medoid to swap out is $j^* := \arg\min_{j \in [k]} \mu_{i^*,j}$. For another candidate point $i \in [n]$ with $i \neq i^*$, let $\Delta_i := \mu_i - \mu_{i^*}$, and for $i = i^*$, let $\Delta_i := \min_j^{(2)} \mu_{i,j} - \min_j \mu_{i,j}$, where $\min_j^{(2)}$ denotes the second smallest value over the indices $j$. To state the following results, we will assume that, for a fixed candidate point $i$ and a randomly sampled reference point $x_s$, the random variable $f(D_i(x_s), \mathcal{A})$ is $\sigma_i$-sub-Gaussian for some known parameter $\sigma_i$ (which, in practice, can be estimated from the data [28]):

**Theorem 3.** *For $\delta = 1/kn^3$, with probability at least $1 - \frac{2}{n}$, Algorithm 1 returns the optimal swap to perform using a total of $M$ distance computations, where*

$$E[M] \leq 6n + \sum_{i \in [n]} \min \left[ \frac{12}{\Delta_i^2} (\sigma_i + \sigma_{i^*})^2 \log kn + B, 3n \right].$$

Intuitively, Theorem 3 states that with high probability, each SWAP iteration of BanditPAM++ returns the same result as BanditPAM and PAM. Since the BUILD step of BanditPAM++ is the same as the BUILD step of BanditPAM, this implies that BanditPAM++ follows the exact same optimization trajectories as BanditPAM and PAM over the entire course of the algorithm with high probability. We formalize this observation in Theorem 4:

**Theorem 4.** *If BanditPAM++ is run on a dataset $\mathcal{X}$ with $\delta = 1/kn^3$, then it returns the same set of $k$ medoids as PAM with probability $1 - o(1)$. Furthermore, the total number of distance computations $M_{\text{total}}$ required satisfies*

$$E[M_{\text{total}}] = O\left(n \log kn\right).$$

**Note on assumptions:** For Theorem 3, we assumed that the data is generated in a way such that the observations $f(D_i(x_s), \mathcal{A})$ follow a sub-Gaussian distribution. Furthermore, for Theorem 4, we assume that the $\Delta_i$'s are not all close to 0, i.e., that we are not in the degenerate arm setting where many of the swaps are equally optimal, and assume that the $\sigma_i$'s are bounded (we formalize these assumptions in Appendix 2). These assumptions have been found to hold in many real-world datasets [28]; see Section 7 and Appendices 1.1, and 2 for more formal discussions.

Additionally, we assume that both BanditPAM and BanditPAM++ place a hard constraint $T$ on the maximum number of SWAP iterations that are allowed. While the limit on the maximum number of swap steps $T$ may seem restrictive, it is not uncommon to place a maximum number of iterations on iterative algorithms. Furthermore, $T$ has been observed empirically to be $O(k)$ [25], consistent with our experiments in Section 6 and Appendix 3.

We note that statements similar to Theorems 3 and 4 can be proven for other values of $\delta$. We provide additional experiments to understand the effects of the hyperparameters $T$ and $\delta$ in Appendix 3.

**Complexity in $k$:** The original BanditPAM algorithm scales as $O(kc(k)n \log kn)$, where $c(k)$ is a problem-dependent function of $k$. Intuitively, $c(k)$ governs the "hardness" of the problem as a function of $k$; as more medoids are added, the average distance from each point to its closest medoid will decrease and the arm gaps (the $\Delta_i$'s) will decrease, increasing the sample complexity in Theorem 3. With the VA technique, BanditPAM++ removes the explicit factor of $k$: each SWAP iteration has complexity $O(c(k)n \log kn)$. The implicit dependence on $k$ may still enter through the term $c(k)$ for a fixed dataset, which we observe in our experiments in Section 6.

## 6 Empirical Results

**Setup:** BanditPAM++ consists of two improvements upon the original BanditPAM algorithm: the VA and PIC techniques. We measure the gains of each technique by presenting an ablation study in which we compare the original BanditPAM algorithm (BP), BanditPAM with only the VA technique (BP+VA), BanditPAM with only the PIC (BP+PIC), and BanditPAM with both the VA and PIC techniques (BP++, the final BanditPAM++ algorithm).

First, we demonstrate that all algorithms achieve the same clustering solution and loss as BanditPAM across a variety of datasets and dataset sizes. In particular, this implies that BanditPAM++ matches

| Dataset Size ($n$): | 10,000 | 15,000 | 20,000 | 25,000 | 30,000 |
|---|---|---|---|---|---|
| MNIST ($L_2, k = 10$) | 1.00 | 1.00 | 1.00 | 1.00 | 1.00 |
| CIFAR10 ($L_1, k = 10$) | 1.00 | 1.00 | 1.00 | 1.00 | 1.00 |
| 20 Newsgroups (cosine, $k = 5$) | 1.00 | 1.00 | 1.00 | 1.00 | 1.00 |

Table 2: Clustering loss of BanditPAM++, normalized to clustering loss of BanditPAM, across a variety of datasets, metrics, and dataset sizes. In all scenarios, BanditPAM++ matches the loss of BanditPAM (in fact, returns the exact same solution).

prior state-of-the-art in clustering quality. Next, we investigate the scaling of all four algorithms in both $n$ and $k$ across a variety of datasets and metrics. We present our results in both sample-complexity and wall-clock runtime. BanditPAM++ outperforms BanditPAM by up to $10\times$. Furthermore, our results demonstrate that each of the VA and PIC techniques improves the runtime of the original BanditPAM algorithm.

For an experiment on a dataset of size $n$, we sampled $n$ datapoints from the original dataset with replacement. In all experiments using the PIC technique, we allowed the algorithm to store up to $1,000$ distance computations per point. For the wall-clock runtime and sample complexity metrics, we divide the result of each experiment by the number of swap iterations $+1$, where the $+1$ accounts for the complexity of the BUILD step.

**Datasets:** We conduct experiments on several public, real-world datasets to evaluate BanditPAM++'s performance: the MNIST dataset, the CIFAR10 dataset, and the 20 Newsgroups dataset. The MNIST dataset [13] contains 70,000 black-and-white images of handwritten digits. The CIFAR10 dataset [12] comprises 60,000 images, where each image consists of $32 \times 32$ pixels and each pixel has 3 colors. The 20 Newsgroups dataset [19] consist of approximately 18,000 posts on 20 topics split in two subsets: train and test. We used a fixed subsample of 10,000 training posts and embedding them into 385-dimensional vectors using a sentence transformer from HuggingFace [7]. We use the $L_2$, $L_1$, and cosine distances across the MNIST, CIFAR10, and 20 Newsgroups datasets, respectively.

### 6.1 Clustering/loss quality

First, we assess the solution quality all four algorithms across various datasets, metrics, and dataset sizes. Table 2 shows the relative losses of BanditPAM++ with respect to the loss of BanditPAM; the results for BP+PIC and BP+VA are identical and omitted for clarity. All four algorithms return identical solutions; this demonstrates that neither the VA nor the PIC technique affect solution quality. In particular, this implies that BanditPAM++ matches the prior state-of-the-art algorithms, BanditPAM and PAM, in clustering quality.

### 6.2 Scaling with $k$

Figure 1 compares the wall-clock runtime scaling with $k$ of BP, BP+PIC, BP+VA, and BP++ on same datasets as Figure 1. Across all data subset sizes, metrics, and values of $k$, BP++ outperforms each of BP+VA and BP+PIC, both of which in turn outperform BP. As $k$ increases, the performance gap between algorithms using the VA technique and the other algorithms increases. For example, on the CIFAR10 dataset with $k = 15$, BanditPAM++ is over $10\times$ faster than BanditPAM. This provides empirical evidence for our claims in Section 5 that the VA technique improves the scaling of the BanditPAM algorithm with $k$.

We provide similar experiments that demonstrate the scaling with $n$ of BanditPAM++ and each of the baseline algorithms in Appendix 3. The results are qualitiatively similar to those shown here; in particular, BanditPAM++ outperforms BP+PIC and BP+VA, which both outperform the original BanditPAM algorithm.

## 7 Conclusions and Limitations

We proposed BanditPAM++, an improvement upon BanditPAM that produces state-of-the-art results for the $k$-medoids problem. BanditPAM++ improves upon BanditPAM using the Virtual Arms

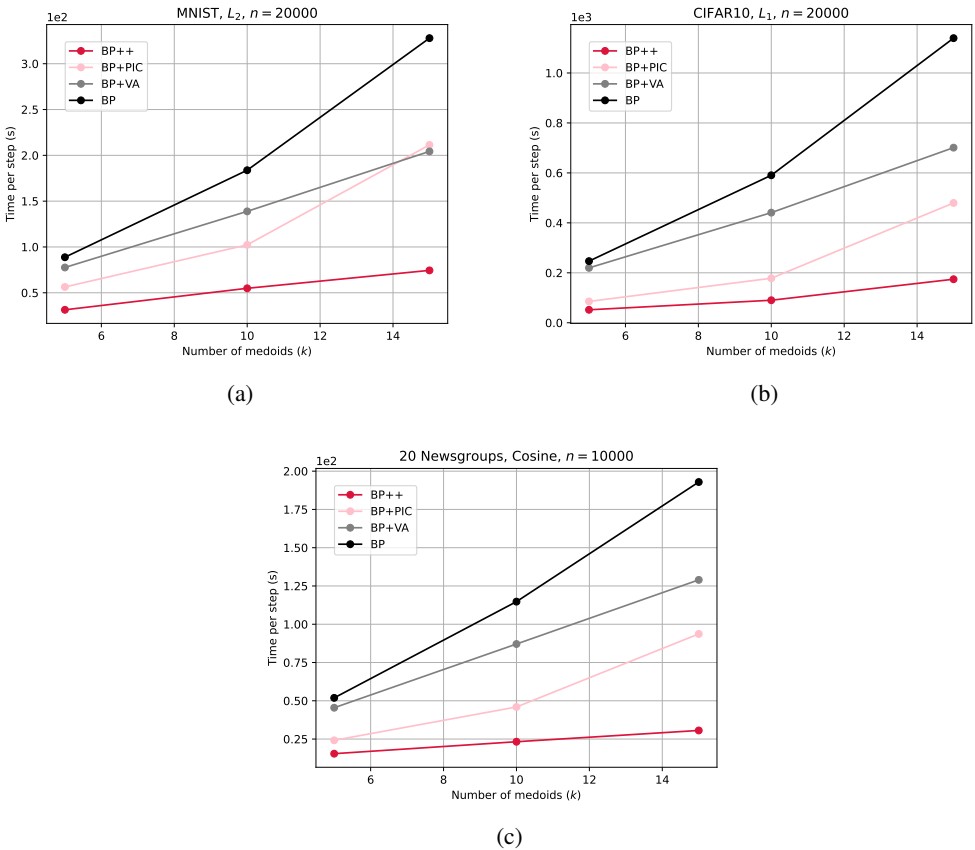

Figure 1: Average wall-clock runtime versus $k$ for various dataset sizes, metrics, and subsample sizes $n$ BP++ outperforms BP+PIC and BP+VA, both of which outperform BP. Negligible error bars are omitted for clarity.

technique, which improves the complexity of each SWAP iteration by $O(k)$, and the Permutation-Invariant Cache, which allows the reuse of computation across different phases of the algorithm. We prove that BanditPAM++ returns the same results as BanditPAM (and therefore PAM) with high probability. Furthermore, our experimental evidence demonstrates the superiority of BanditPAM++ over baseline algorithms; across a variety of datasets, BanditPAM++ is up to $10\times$ faster than prior state-of-the-art while returning the same results. While the assumptions of BanditPAM and BanditPAM++ are likely to hold in many practical scenarios, it is important to acknowledge that these assumptions can impose limitations on our approach. Specifically, when numerous arm gaps are narrow, BanditPAM++ might employ a naïve and fall back to brute force computation. Similarly, if the distributional assumptions on arm returns are violated, the complexity of BanditPAM++ may be no better than PAM. We discuss these settings in greater detail in Appendix 1.1.

## Acknowledgements

We would like to thank the anonymous Reviewers, PCs, and ACs for their helpful feedback on our paper. Mo Tiwari was supported by a Stanford Interdisciplinary Graduate Fellowship (SIGF) and a Standard Data Science Scholarship. The work of Ilan Shomorony was supported in part by the National Science Foundation (NSF) under grant CCF-2046991. Martin Zhang is supported by NIH R01 MH115676.

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
