# Appendix

## 1 BanditPAM: Additional Details

In this section, we provide additional information about the original BanditPAM algorithm. Bandit-PAM uses a bandit-based randomized algorithm to cast the BUILD and SWAP steps of the original PAM algorithm in terms of the following equation:

$$\arg\min_{x \in \mathcal{S}_{\text{tar}}} \frac{1}{|\mathcal{S}_{\text{ref}}|} \sum_{x_j \in \mathcal{S}_{\text{ref}}} g_x(x_j) \tag{9}$$

for target points $\mathcal{S}_{\text{tar}}$, reference points $\mathcal{S}_{\text{ref}}$, and an objective function $g_x(\cdot)$ that depends on the target point $x$. Both the BUILD and SWAP problems can be written as instances of Problem (9) with:

$$\text{BUILD: } \mathcal{S}_{\text{tar}} = \mathcal{X} \setminus \mathcal{M}_l, \ \mathcal{S}_{\text{ref}} = \mathcal{X}, \ g_x(x_j) = \left( d(x, x_j) - \min_{m' \in \mathcal{M}_l} d(m', x_j) \right) \wedge 0. \tag{10}$$

$$\text{SWAP: } \mathcal{S}_{\text{tar}} = \mathcal{M} \times (\mathcal{X} \setminus \mathcal{M}), \ \mathcal{S}_{\text{ref}} = \mathcal{X}, \ g_x(x_j) = \left( d(x, x_j) - \min_{m' \in \mathcal{M} \setminus \{m\}} d(m', x_j) \right) \wedge 0. \tag{11}$$

where $\mathcal{M}_l$ is the current set of $l \leq k$ medoids.

Crucially, in the SWAP search, each *pair* of medoid-and-nonmedoid points $(m, x)$ is treated as one target point (i.e. arm) in $\mathcal{S}_{\text{tar}}$ in the original BanditPAM formulation. The `Adaptive-Search` algorithm, Algorithm 2, is then used for best-arm identification (finding the optimal element of $\mathcal{S}_{\text{tar}}$) for each of the BUILD and SWAP steps.

We also present the `Adaptive-Search` algorithm, Algorithm 2, specialized for the SWAP step explicitly in Algorithm 3, for ease of comparison to Algorithm 1. We note the differences between the SWAP step of BanditPAM (Algorithm 3) and the SWAP step of BanditPAM++ (Algorithm 1). Whereas the original BanditPAM algorithm considers each medoid-nonmedoid pair as an arm in the best-arm identification problem, BanditPAM++ only considers each nonmedoid as an arm. (This change, in turn, allows BanditPAM++ to be formulated as a SPIMAB.) Furthermore, whereas the original BanditPAM algorithm may compute $k$ distances to update all $k$ arms for a fixed nonmedoid (Lines 6-8 of Algorithm 3), BanditPAM++ only computes a maximum of three distances (Lines 8-10 of Algorithm 1). This is what removes the explicit dependence on $k$ from the computational complexity of BanditPAM++, as discussed in Section 5.

---

**Algorithm 2** `Adaptive-Search` ( $\mathcal{S}_{\text{tar}}, \mathcal{S}_{\text{ref}}, g_x(\cdot), \delta, \sigma_x$ )

---

1: $\mathcal{S}_{\text{solution}} \leftarrow \mathcal{S}_{\text{tar}}$            ▷ Set of potential solutions to Problem (9)
2: $t' \leftarrow 0$                ▷ Number of reference points evaluated
3: For all $x \in \mathcal{S}_{\text{tar}}$, set $\hat{\mu}_x \leftarrow 0, C_x \leftarrow \infty$    ▷ Initial mean and confidence interval for each arm
4: **while** $t' < |\mathcal{S}_{\text{ref}}|$ and $|\mathcal{S}_{\text{solution}}| > 1$ **do**
5:    Draw a reference point $s$ from $\mathcal{S}_{\text{ref}}$
6:    **for all** $x \in \mathcal{S}_{\text{solution}}$ **do**
7:      $\hat{\mu}_x \leftarrow \frac{t' \hat{\mu}_x + g_x(s)}{t'+1}$           ▷ Update running mean
8:      $C_x \leftarrow \sigma_x \sqrt{\frac{\log(\frac{1}{\delta})}{t'+1}}$         ▷ Update confidence interval
9:      $\mathcal{S}_{\text{solution}} \leftarrow \{x : \hat{\mu}_x - C_x \leq \min_y(\hat{\mu}_y + C_y)\}$    ▷ Remove suboptimal points
10:      $t' \leftarrow t' + 1$
11: **if** $|\mathcal{S}_{\text{solution}}| = 1$ **then**
12:    **return** $x^* \in \mathcal{S}_{\text{solution}}$
13: **else**
14:    Compute $\mu_x$ exactly for all $x \in \mathcal{S}_{\text{solution}}$
15:    **return** $x^* = \arg\min_{x \in \mathcal{S}_{\text{solution}}} \mu_x$

---

---

**Algorithm 3** BanditPAM SWAP Step ( $f_j(D_j, a_1, \ldots, a_k), \delta, \sigma_x,$ )

---

1:  $\mathcal{S}_{\text{solution}} \leftarrow [n] \times [k] = \{(1,1), \ldots, (1,k), (2,1), \ldots, (n,k)\}$   ▷ Potential swaps
2:  $t' \leftarrow 0$   ▷ Number of reference points evaluated
3:  For all $(i,j) \in \mathcal{S}_{\text{tar}}$, set $\hat{\mu}_{i,j} \leftarrow 0, C_{i,j} \leftarrow \infty$   ▷ Initial mean and confidence intervals
4:  **while** $t' < |\mathcal{S}_{\text{ref}}|$ and $|\mathcal{S}_{\text{solution}}| > 1$ **do**
5:      Draw the next sample $x_s$ from $[n]$ randomly
6:      **for all** $(i,j) \in \mathcal{S}_{\text{solution}}$ **do**
7:          $\hat{\mu}_{i,j} \leftarrow \frac{t'\hat{\mu}_{i,j} + f(D_i(x_s), a_1, \ldots, a_k)}{t'+1}$   ▷ Update running mean; special case when $j$ is $c(s)$
8:          $C_{i,j} \leftarrow \sigma_{i,j}\sqrt{\frac{\log(\frac{1}{\delta})}{t'+1}}$   ▷ Update confidence interval
9:          $\mathcal{S}_{\text{solution}} \leftarrow \{i, j : \hat{\mu}_{i,j} - C_{i,j} \leq \min_{i,j}(\hat{\mu}_{i,j} + C_{i,j})\}$   ▷ Remove suboptimal swaps
10:         $t' \leftarrow t' + 1$
11: **if** $|\mathcal{S}_{\text{solution}}| = 1$ **then**
12:     **return** $(i^*, j^*) \in \mathcal{S}_{\text{solution}}$
13: **else**
14:     Compute $\mu_{i,j}$ exactly for all $(i^*, j^*) \in \mathcal{S}_{\text{solution}}$
15:     **return** $(i^*, j^*) = \arg\min_{(i,j) \in \mathcal{S}_{\text{solution}}} \mu_{(i,j)}$

---

### 1.1 Distributional assumptions and violations

For BanditPAM++ to have computational gains over the original PAM algorithm, several distributional assumptions must be met. First, we assume that the observations $R_i^t$ are $\sigma_{i,t}$-sub-Gaussian for some values of $\sigma_{i,t}$. Intuitively, this implies that the samples we observe for $R_i^t$ are representative of their true mean and that the sample mean concentrates about its true mean by Hoeffding's inequality. This assumption holds, for example, for any bounded dataset and has been found to hold in many real-world datasets [28].

Another assumption of BanditPAM++ that is made to achieve $O(n \log n)$ complexity is about the distribution of arm means and gaps (the $\Delta_i$'s). More specifically, we assume we are never in the degenerate setting where all $\Delta_i$'s are 0, i.e., where all potential swaps are equally good. Reasonable distributions of the $\Delta_i$'s are often observed in practice; for examples and a more formal discussion, we refer the reader to [28] and references therein.

Finally, we also assume that there is a hard limit $T$ on the number of swaps performed. This is a common restriction in iterative algorithms and empirically has not been found to significantly degrade clustering results for $T = O(k)$. In our additional experiments in Appendix 3, we empirically validate this observation in several settings.

We emphasize that the same distributional assumptions are made by both BanditPAM and Bandit-PAM++. More specifically, both algorithms will demonstrate superiority over PAM under exactly the same, rather general conditions. Furthermore, under those conditions, BanditPAM++ will generally outperform BanditPAM.

## 2 Proofs of Theorems 1, 2, 3 and 4

In this Appendix, we provide proofs for Theorems 1, 2, 3, and 4.

**Theorem 1.** *Let $\Delta l_{m,x_i}(x_s)$ be the change in clustering loss induced on point $x_s$ by swapping medoid $m$ with nonmedoid $x_i$, given in Equation (7), with $x_s$ and $x_i$ fixed. Then the values $\Delta l_{m,x_i}(x_s)$ for $m = 1, \ldots, k$ are equal, except possibly where $m$ is the medoid for reference point $x_s$.*

*Proof.* Consider the effect of swapping medoid $m$ with nonmedoid $x_i$ on point $x_s$. To compute $\Delta l_{m,x_i}(x_s)$, we must consider four possible cases depending on whether point $x_s$ was assigned to medoid $m$ as its closest medoid before the swap and whether $x_s$ would be assigned to medoid $x_i$ after the swap. We denote by $m_{c(s)}$ the medoid $x_s$ is assigned to before the swap, and by $m_{c^{(2)}(s)}$ the second-closest medoid to $x_s$ before the swap.

Case 1: $m$ is the current medoid for reference point $x_s$, i.e., $m = m_{c(s)}$, and $x_i$ would become the medoid for $x_s$ after the swap. Then $\Delta l_{m,x_i}(x_s) = d(x_i, x_s) - d(m, x_s) = d(x_i, x_s) - d(m_{c(s)}, x_s)$.

Case 2: $m$ is *not* the current medoid for reference point $x_s$, i.e., $m \neq m_{c(s)}$, and $x_i$ would become the medoid for $x_s$ after the swap. Then $\Delta l_{m,x_i}(x_s) = d(x_i, x_s) - d(m_{c(s)}, x_s)$.

Case 3: $m$ is the current medoid for reference point $x_s$, i.e., $m = m_{c(s)}$, and $x_i$ would *not* become the medoid for $x_s$ after the swap. Then $\Delta l_{m,x_i}(x_s) = d(m_{c^{(2)}(s)}, x_s) - d(m, x_s) > 0$.

Case 4: $m$ is *not* the current medoid for reference point $x_s$, i.e., $m \neq m_{c(s)}$, and $x_i$ would *not* become the medoid for $x_s$ after the swap. Then $\Delta l_{m,x_i}(x_s) = 0$.

We can condense these four cases into a single expression as:

$$\Delta l_{m,x_i}(x_s) = -d(m_{c(s)}, x_s) + \mathbb{1}_{x_s \notin \mathcal{C}_m} \min[d(m_{c(s)}, x_s), d(x_i, x_s)] + \tag{12}$$

$$\mathbb{1}_{x_s \in \mathcal{C}_m} \min[d(m_{c^{(2)}(s)}, x_s), d(x_i, x_s)] \tag{13}$$

where $\mathcal{C}_m$ denotes the set of points whose closest medoid is $m$ and $d(m_{c(s)}, x_s)$ and $d(m_{c^{(2)}(s)}, x_s)$ are the distance from $x_s$ to its nearest and second nearest medoid, respectively, before the swap is performed.

Note that Equation 12 depends only on $m$ via the terms $\mathbb{1}_{x_s \in \mathcal{C}_m}$ and $\mathbb{1}_{x_s \notin \mathcal{C}_m}$, so for the $k-1$ values of $m$ for which $x_s \notin \mathcal{C}_m$, we must have that $\Delta l_{m,x_i}(x_s)$ is equal (for fixed $x_i$ and $x_s$). $\qquad\square$

**Theorem 2.** *Let $\mathcal{X} = \{x_1, \ldots, x_S\}$ be the reference points of $D_i$, and let $\pi$ be a random permutation of $\{1, \ldots, S\}$. Then for any $c \leq S$, $\sum_{q=1}^c D_i(x_{\pi(p_q)})$ has the same distribution as $\sum_{q=1}^c D_i(x_{p_q})$, where each $p_q$ is drawn uniformly without replacement from $\{1, \ldots, S\}$.*

*Proof.* Since $\pi$ is a permutation drawn uniformly at random over the set of possible permutations, the probability that any integer $p_q$ appears in the first $c$ elements of the ordered sequence $\{\pi(1), \ldots, \pi(S)\}$ is $\frac{c}{k}$. This is the same as the probability that any integer $p_q$ occurs in the first $c$ elements of the ordered sequence $\{1, \ldots, S\}$. This, in turn, implies that $\pi(p_q)$ and $p_q$ have the same distribution for all $q$. Since the indices $\pi(p_q)$ and $p_q$ have the same distribution, so do the term-wise elements of each of the sums $\sum_{q=1}^c D_i(x_{\pi(p_q)})$ and $\sum_{q=1}^c D_i(x_{p_q})$.

$\qquad\square$

The proofs of Theorems 3 and 4 are similar to those for the original BanditPAM algorithm; however, they must now be adapted to the modified algorithm that uses Theorem 1 to change the best-arm identification problem.

**Theorem 3.** *For $\delta = 1/kn^3$, with probability at least $1 - \frac{2}{n}$, Algorithm 1 returns the optimal swap to perform using a total of $M$ distance computations, where*

$$E[M] \leq 6n + \sum_{i \in [n]} \min \left[ \frac{12}{\Delta_i^2} (\sigma_i + \sigma_{i^*})^2 \log kn + B, 3n \right].$$

*Proof.* The proof is similar to that of the original BanditPAM algorithm [28]; however, it must be adapted for the Virtual Arms technique.

First, we show that, with probability at least $1 - \frac{2}{n}$, all confidence intervals computed throughout the algorithm are true confidence intervals, in the sense that they contain the true parameters $\mu_{i,j}$. To see this, notice that for a fixed $i, j$ and a fixed iteration of the algorithm, $\hat{\mu}_{i,j}$ is the average of $t$ i.i.d. samples of a $\sigma_i$-sub-Gaussian distribution. From Hoeffding's inequality,

$$\Pr\left(|\mu_{i,j} - \hat{\mu}_{i,j}| > C_{i,j}\right) \leq 2\exp\left(-\frac{tC_{i,j}^2}{2\sigma_i^2}\right) =: 2\delta.$$

where we used $\mathbb{E}[\hat{\mu_{i,j}}] = \mu_{i,j}$ by Theorem 2.

Note that there are at most $kn^2$ such confidence intervals computed across all arms and all steps of the algorithm. With $\delta = 1/kn^3$, we see that $\mu_{i,j} \in [\hat{\mu}_{i,j} - C_{i,j}, \hat{\mu}_{i,j} + C_{i,j}]$ for every $i, j$ and for every step of the algorithm with probability at least $1 - \frac{2}{n}$, by the union bound over at most $kn^2$ confidence intervals.

Next, we prove the correctness of Algorithm 1. Let $(i^*, j^*) = \arg\min_{(i,j)} \mu_{i,j}$ be the desired output of the algorithm. First, observe that the main `while` loop in the algorithm can only run $n$ times, so the algorithm must terminate. Furthermore, if all confidence intervals throughout the algorithm are correct, it is impossible for $(i^*, j^*)$ to be removed from the set of candidate target points. Hence, $(i^*, j^*)$ (or some $(i', j')$ with $\mu_{i',j'} = \mu_{i^*,j^*}$) must be returned upon termination with probability at least $1 - \frac{2}{n}$.

Finally, we consider the complexity of Algorithm 1. Let $t$ be the total number of arm pulls computed for each of the arms remaining in the set of candidate arms at some point in the algorithm. As in Section 5, let $\mu_i := \min_{j \in [k]} \mu_{i,j}$, let $i^* := \arg\min_{i \in [n]} \mu_i$ be the optimal point to swap in to the set of medoids, so that the medoid to swap out is $j^* := \arg\min_{j \in [k]} \mu_{i^*,j}$, and for another candidate point $i \in [n]$ with $i \neq i^*$, let $\Delta_i := \mu_i - \mu_{i^*}$. Furthermore, for $i = i^*$, let $\Delta_i := \min_j^{(2)} \mu_{i,j} - \min_j \mu_{i,j}$, where $\min_j^{(2)}$ denotes the second largest value over the indices $j$. Notice that, for any suboptimal arm $(i, j) \neq (i^*, j^*)$, we must have $C_{i,j} = \sigma_i \sqrt{\log(\frac{1}{\delta})/t}$. With $\delta = 1/kn^3$ as above and $\Delta_{i,j} := \mu_{i,j} - \mu_{i^*,j^*}$, if $t > \frac{12}{\Delta_i^2} (\sigma_i + \sigma_{i^*})^2 \log kn$, then

$$2(C_{i,j} + C_{i^*,j^*}) = 2(\sigma_i + \sigma_{i^*}) \sqrt{\log(kn^3)/t} \tag{14}$$
$$< \Delta_i \tag{15}$$
$$:= \min_{j'} \Delta_{i,j'} \tag{16}$$
$$\leq \Delta_{i,j} \tag{17}$$
$$= \mu_{i,j} - \mu_{i^*,j^*}, \tag{18}$$

for all $j$, and so

$$\hat{\mu}_{i,j} - C_{i,j} > \mu_{i,j} - 2C_{i,j}$$
$$= \mu_{i^*,j^*} + \Delta_{i,j} - 2C_{i,j}$$
$$\geq \mu_{i^*,j^*} + 2C_{i^*,j^*}$$
$$> \hat{\mu}_{i^*,j^*} + C_{i^*,j^*}$$

implying that $(i, j)$ must be removed from the set of candidate arms by iteration $t$. Hence, the number of distance computations $M_i$ required to exclude *all* arms $(i, j)$ where $i \neq i^*$, or for all $(i, j)$ with $i = i^*$ but $j \neq j^*$, is at most

$$M_i \leq \min\left[ \frac{12}{\Delta_i^2} (\sigma_i + \sigma_{i^*})^2 \log kn + 1, 3n \right].$$

Notice that this holds simultaneously for all $(i, j)$ with probability at least $1 - \frac{2}{n}$. We conclude that the total number of distance computations $M$ satisfies

$$E[M] \leq E[M \mid \text{all confidence intervals are correct}] + \frac{2}{n}(3n^2)$$
$$\leq 6n + \sum_{i \in [n]} \min\left[ \frac{12}{\Delta_i^2} (\sigma_i + \sigma_{i^*})^2 \log kn + 1, 3n \right]$$

where we used the fact that the maximum number of distance computations per target point is $3n$. $\quad\square$

**Theorem 4.** *If BanditPAM++ is run on a dataset $\mathcal{X}$ with $\delta = 1/kn^3$, then it returns the same set of $k$ medoids as PAM with probability $1 - o(1)$. Furthermore, the total number of distance computations $M_{\text{total}}$ required satisfies*

$$E[M_{\text{total}}] = O(n \log kn).$$

*Proof.* This proof is similar to that for the original BanditPAM algorithm [28]. Since the BUILD step of BanditPAM++ is the same as the BUILD step of the original BanditPAM algorithm, it suffices to use the fact from Theorem 3 that each SWAP step of BanditPAM++ agrees with each SWAP step of BanditPAM and PAM.

From Theorem 3, the probability that Algorithm 1 does not return the target point $i, j$ with the smallest value of $\mu_{i,j}$ in a single call, i.e. that the result of Algorithm 1 will differ from the corresponding step in PAM, is at most $2/n$. By the union bound over all $T$ calls to Algorithm 1, the probability that BanditPAM++ does not return the same set of $k$ medoids as PAM is at most $2T/n = o(1)$, since $T$ is taken a predefined constant. This proves the first claim of Theorem 4.

It remains to show that $E[M_{\text{total}}] = O(n \log kn)$. Note that, if a random variable is $\sigma$-sub-Gaussian, it is also $\sigma'$-sub-Gaussian for $\sigma' > \sigma$. Hence, if we have a universal upper bound $\sigma_{\text{ub}} > \sigma_i$ for all $i, j$, Algorithm 1 can be run with $\sigma_{\text{ub}}$ replacing each $\sigma_i$. In that case, a direct consequence of Theorem 3 is that the total number of distance computations per call to Algorithm 1 satisfies

$$E[M] \leq 6n + \sum_{i \in [n]} 48 \frac{\sigma_{\text{ub}}^2}{\Delta_i^2} \log kn + 1 \leq 6n + 48 \left( \frac{\sigma_{\text{ub}}}{\min_i \Delta_i} \right)^2 n \log kn. \tag{19}$$

Furthermore, as proven in Appendix 2 of [4], such an instance-wise bound, which depends on the $\Delta_i$'s, converts to an $O(n \log kn)$ bound when the $\mu_{i,j}$'s follow a sub-Gaussian distribution. Moreover, since at most $T$ calls to Algorithm 1 are made, from (19) we see that the total number of distance computations $M_{\text{total}}$ required by BanditPAM++ satisfies $E[M_{\text{total}}] = O(n \log kn)$. $\qquad\square$

## 3 Additional Experiments

In this appendix, we provide additional experiments, including results moved here from the main text due to space constraints. First, we present the wall-clock runtimes and sample complexities of various algorithms, including BanditPAM++, across a variety of datasets and metrics, for different dataset sizes $n$. We then show that BanditPAM++ is not sensitive to the choice of error probability $\delta$. We then demonstrate that the loss of various real-world problems stops decreasing after approximately $T = k$ swaps, as has been observed in prior work [28, 25]. Finally, we provide a breakdown of the speedups across both the BUILD and SWAP steps of BanditPAM++ over BanditPAM.

### 3.1 Scaling with $n$

Appendix Figure 1 compares the wall-clock runtime scaling with $n$ of BP, BP+PIC, BP+VA, and BP++ on the MNIST ($L_2$ distance, $k = 10$), CIFAR10 ($L_1$ distance, $k = 10$), and 20 Newsgroups (cosine distance, $k = 5$) datasets. Across all data subset sizes, metrics, and values of $k$, BP++ outperforms each of BP+VA and BP+PIC, both of which in turn outperform BP.

We also show the results of the same experiments as in Appendix 1, but where we measure sample complexity (number of distance computations) instead of wall-clock runtime, in Appendix Figure 2. Appendix Figure 2 compares the sample complexity versus data subsample size $n$ of each algorithm for the same experimental settings as in Appendix Figure 1. In these experiments, the cost of a distance computation is set to $1$ and the cost of a cache hit is set to $0$. The results are qualitatively similar to those in Appendix Figure 1.

### 3.2 Loss of BanditPAM++ with varying $\delta$

We now demonstrate that BanditPAM++ is not sensitive to the choice of error probability hyperparameter $\delta$. Appendix Table 1 shows the loss of BanditPAM++, normalized to the loss of BanditPAM, across various values of $\delta$. The original BanditPAM algorithm has been shown to be insensitive to $\delta$ and agrees with PAM [28] Therefore, Appendix Table 1 implies that BanditPAM++ agrees with PAM as well.

### 3.3 Loss of BanditPAM++ with varying $T$

We now demonstrate that the choice of maximum swap steps $T = k$ is reasonable in practice. It has been observed in prior work that after $k$ swap steps, the clustering loss (Equation 1) does not significantly decrease [28, 25]. Appendix Figure 3 demonstrates the clustering loss of BanditPAM++ as $T$ increases. Beyond $T = k$, the clustering loss decreases much more slowly.

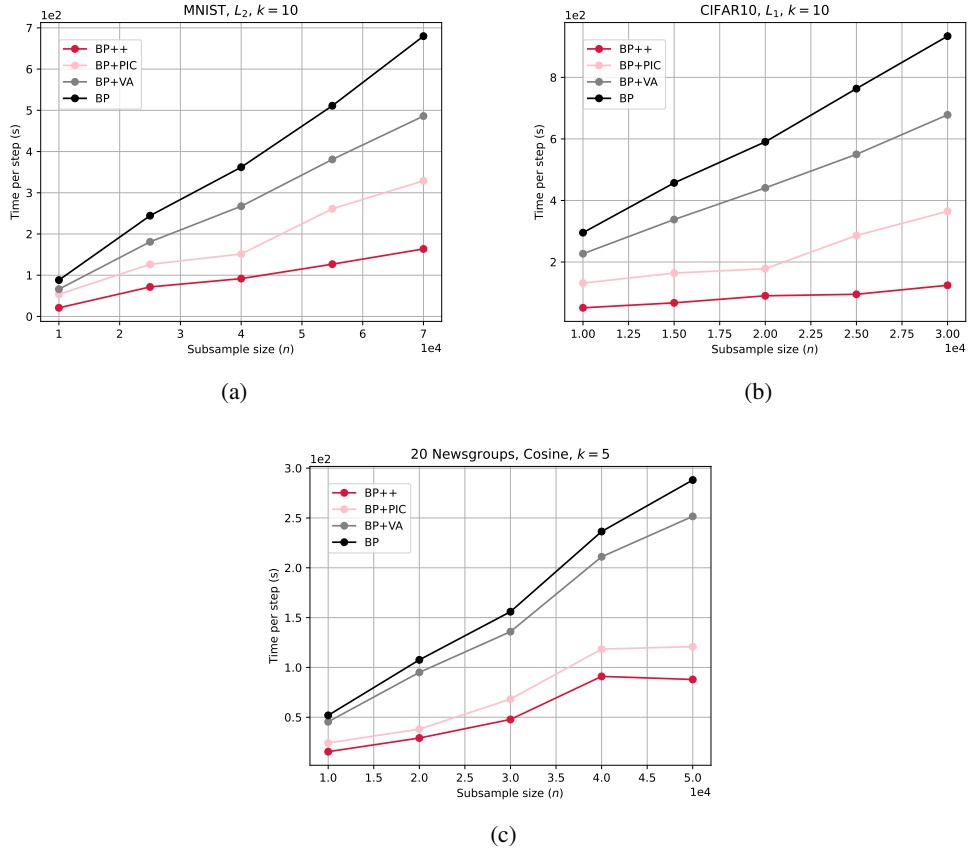

Appendix Figure 1: Average wall-clock runtime versus dataset size $n$ for various dataset sizes, metrics, and values of $k$. BP++ outperforms BP+PIC and BP+VA, both of which outperform BP. Negligible error bars are omitted for clarity.

| Dataset | $10^{-2}$ | $10^{-3}$ | $10^{-5}$ | $10^{-10}$ |
|---|---|---|---|---|
| MNIST ($L_2, k = 10$) | 1.00 | 1.00 | 1.00 | 1.00 |
| CIFAR10 ($L_1, k = 10$) | 1.00 | 1.00 | 1.00 | 1.00 |
| 20 Newsgroups (Cosine, $k = 5$) | 1.00 | 1.00 | 1.00 | 1.00 |

Appendix Table 1: Loss of BanditPAM++, normalized to loss of BanditPAM, with $\delta$ ranging from $10^{-2}$, $10^{-3}$, $10^{-5}$, and $10^{-10}$. BanditPAM++ has the exact same clustering loss with BanditPAM (and therefore PAM) for various values $\delta$.

### 3.4 Speedups of BUILD and SWAP steps

We now present the average speedups for both the BUILD and SWAP steps for BanditPAM++ compared to the original BanditPAM algorithm, to separately assess the impacts of the Virtual Arms and Permutation-Invariant Caching techniques. Whereas the VA only improves the SWAP step, the PIC improves both the BUILD and SWAP steps.

In Appendix Table 2, we present the wall-clock speedup of BanditPAM++ compared to BanditPAM across the MNIST, CIFAR10, and 20 Newsgroups datasets described in Section 6. BanditPAM++ shows significant gains over BanditPAM in both phases of the algorithm, with the gains in the SWAP phase more pronounced.

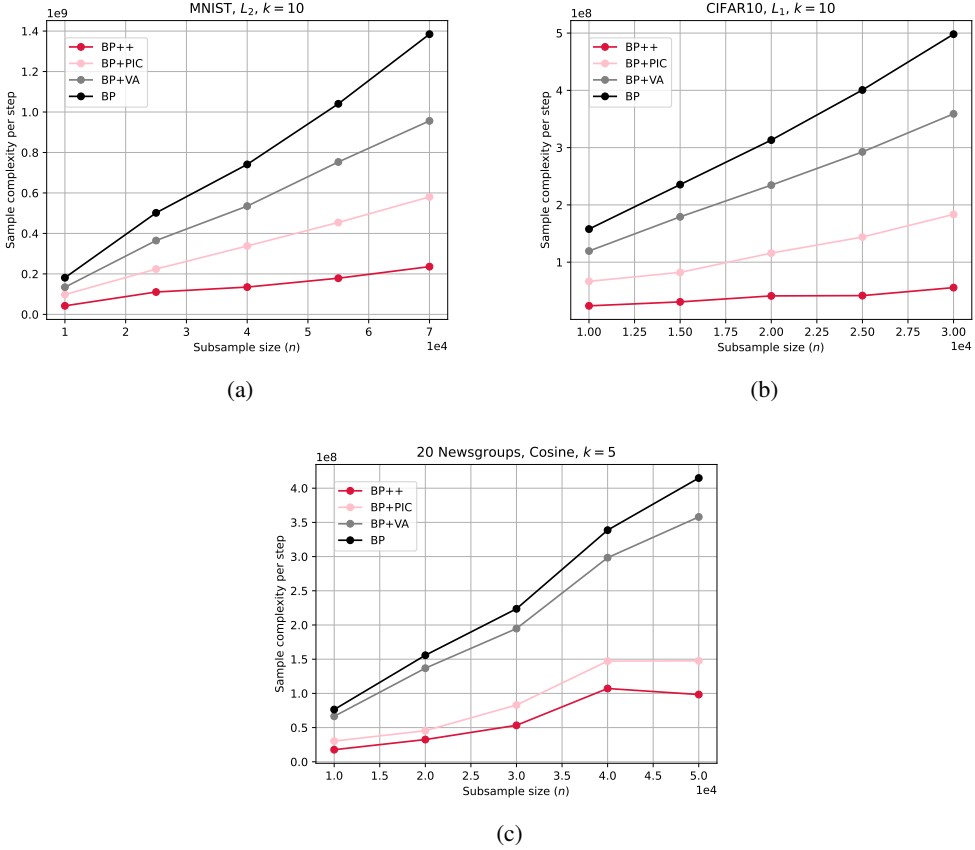

(a)

(b)

(c)

Appendix Figure 2: Average sample complexity versus dataset size $n$ for various dataset sizes, metrics, and values of $k$. BP++ outperforms BP+PIC and BP+VA, both of which outperform BP. Negligible error bars are omitted for clarity.

| Dataset | BUILD + SWAP | SWAP only | BUILD only |
|---|---|---|---|
| MNIST | $\times 3.07$ \| $\times 4.21$ \| $\times 4.15$ | $\times 3.80$ \| $\times 6.38$ \| $\times 5.15$ | $\times 1.36$ \| $\times 1.75$ \| $\times 1.44$ |
| CIFAR | $\times 5.45$ \| $\times 5.77$ \| $\times 10.22$ | $\times 8.14$ \| $\times 6.97$ \| $\times 10.51$ | $\times 1.92$ \| $\times 2.69$ \| $\times 2.52$ |
| 20 Newsgroups | $\times 2.69$ \| $\times 4.94$ \| $\times 6.93$ | $\times 4.85$ \| $\times 6.68$ \| $\times 9.07$ | $\times 1.59$ \| $\times 1.75$ \| $\times 1.86$ |

Appendix Table 2: Average Runtime Speedup Summary: Wall-clock speedup of BanditPAM++ compared to BanditPAM on the four datasets MNIST, CIFAR10, and 20 Newsgroups. Results are for $n = 10,000$ for the BUILD and SWAP weighted average, SWAP phase only, and BUILD phase only. The BUILD phase only leverages permutation-invariant caching, whereas the other two settings also leverage Virtual Arms. The three speedup values in each cell correspond to experiments where $k = 5, 10$, and $15$ respectively.

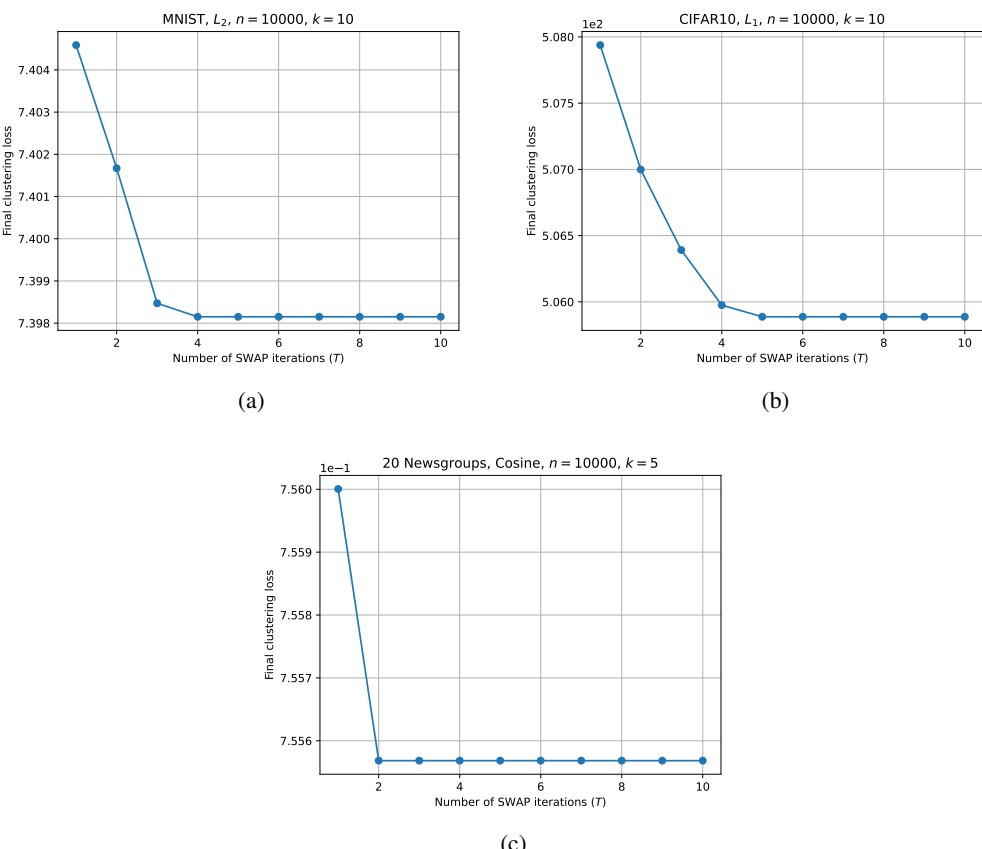

Appendix Figure 3: Clustering loss versus maximum number of SWAP iterations, $T$, for the MNIST, CIFAR10, and 20 Newsgroups datasets for various values of $k$. Beyond $T = k$, the loss shows very little change. BanditPAM++ and BanditPAM have the same loss for all $T$ and track the same optimization trajectory.