# OpenReview forum: "BanditPAM++: Faster $k$-medoids Clustering"
_NeurIPS.cc/2023/Conference — NeurIPS 2023 poster_

### Official Review · Reviewer_mktW · 2023-06-25

**Soundness:** 2 fair
**Presentation:** 3 good
**Contribution:** 2 fair
**Rating:** 6
**Confidence:** 2

**Summary:**

This paper studies a clustering problem (k-medoid) in which we have to select k centers among the datapoints in order to minimize the sum of distances of points to their closest center. For this problem, there is an intuitive algorithm which works well empirically (the PAM algorithm): (1) add greedily centers one by one by selected the next center to be the one which maximizes the decrease in cost (BUILD phase), (2) once we have selected k-centers, we try to swap one center for another datapoint again greedily (SWAP phase). We stop after some iterations (usually we stop after O(k) iterations). This algorithm works well in practice but is quite expensive in terms of computation. Therefore, several works have designed variants of the PAM algorithm to lower the computational cost without sacrificing too much of the quality of the solution. BanditPAM is one of these algorithm which returns the same solution as PAM (with high probability) but performs much less distance computations.

This paper further improve BanditPAM to propose BanditPAM++. They show that BanditPAM++ also returns the same solution as PAM with high probability, and uses less comparisons compared to BanditPAM. On some real-world datasets, BanditPAM++ can be up to 10x faster than BanditPAM.


**Strengths:**

I think this is an interesting and useful result. The experimental improvements seem substantial.

**Weaknesses:**

The techniques used to obtain this algorithm did not seem very innovative (I am not an expert in bandit algorithms however).

typos:
Line 118: fundamentally
Line 161: missing Delta I think
Pseudocode page 6: I guess m_r^(2) the second closest medoid ? I am not sure this is defined in this page.


**Questions:**

In the beginning of intro, I got confused by the remark that k-means does not generalize to other metrics. What do you mean by that? Isn’t equation (1) such a generalization?
About the PAM algorithm in general, it seems that there is no worst-case guarantee for this method. Is there any practical algorithm that also gives some kind of worst-case guarantee in terms of clustering cost?

**Limitations:**

yes

---

> ### Author Rebuttal · Authors · 2023-08-09
>
> We thank Reviewer mktW for their thoughtful comments and helpful feedback. We provide a point-by-point response below.
>
> Response to W1: We would like to clarify our contributions and their novelty.
>
> BanditPAM++ improves upon BanditPAM via two novel techniques: the Virtual Arms (VA) technique and Permutation-Invariant Caching (PIC). Both techniques are tailored for BanditPAM: VA relies on the observation that the updates of $k$ medoids for a given candidate point are correlated, and PIC relies on the observation that the sequence of MAB searching problems uses the same set of reference points. Moreover, both techniques bring substantial improvements: an $O(k)$ complexity improvement in the SWAP step by VA and wall-clock time speed up by PIC. Our resulting algorithm, BanditPAM++ can be an order of magnitude faster than BanditPAM in some settings (Table 1 of the additional experiments provided in the "global" response to all reviewers). Since these are not general techniques, and they substantially improve the state-of-the-art $k$-medoids clustering algorithm, we argue that they are non-trivial and significant.
>
> The VA technique is applied to accelerate the SWAP steps in BanditPAM++. Whereas the original BanditPAM algorithm considers all $kn$ possible (medoid, datapoint) pairs as "arms" during each SWAP step, BanditPAM++ considers only each of the $n$ datapoints as "arms". This is emphasized in blue in Line 1 of Algorithm 2. This methodological change is permitted by Theorem 1, which codifies our observation that at least $k-1$ of the arms from the original BanditPAM algorithm are receiving the same update in Line 7 of Algorithm 1. We show that this methodological change in BanditPAM++ results in the same answers as PAM and BanditPAM in Theorems 3 and 4, which means that BanditPAM++ matches state-of-the-art algorithms in clustering quality.
>
> Crucially, the fact that BanditPAM++ now considers every datapoint as an "arm" during each of the BUILD and SWAP steps permits our second observation and the PIC technique. Because the set of objects that are considered "arms" is the same across each phase of BanditPAM++, we can now cache the quantities that are expensive to compute: the distances between datapoints. Choosing a permutation of datapoints a priori and using this permutation across all phases of BanditPAM++ increases the proportion of cache hits when the cache is too small to hold all $n^2$ pairwise distances, as is the case in the setting we are interested in: when $n$ is large. These observations are formalized in the SPIMAB framework of our paper. The Permutation-Invariant Caching technique improves both the BUILD and SWAP procedures of BanditPAM in BanditPAM++.
>
> Response to W2: Thank you for pointing out these typos; we will correct them in the final paper. Yes, $m^{(2)}_r$ is the second-closest medoid.
>
> Response to Q1 (k-means): We meant to emphasize what is written in text above Equation 1, on Line 24; namely, that in Equation 1, $k$-means must use a specific form of distance metric $d(u, x_j)$ for algorithmic efficiency. Usually, this is the squared $L_2$ distance, though other very particular metrics can also be used for $k$-means.
>
> The general point is still that $k$-means typically requires specific distance metrics for efficient solutions. In contrast, $k$-medoids is agnostic to the distance metric being used. In fact, the distance function being used in $k$-medoids need not even be a metric and can be an arbitrary dissimilarity function (i.e., may violate the triangle inequality, may be negative, and may not be symmetric). For this reason, $k$-medoids has been used to cluster "exotic" objects such as trees [20].
>
>
> Response to Q1 (worst-case): One may notice that the optimization objective in Equation 2 (when negated) is actually submodular with respect to the set $\mathcal{M}$. It is a well-known result in submodular optimization theory that greedy optimization of submodular results in a solution that is at worst $1 - \frac{1}{e} \approx 0.63$ of optimal (e.g., [G]).
>
> For this reason, after the BUILD steps of PAM (or BanditPAM or BanditPAM++, since they all follow the same optimization trajectory), the loss of the current set of $k$ medoids is no worse than $\approx 0.63$ of optimal, when the loss is appropriately negated. Since the SWAP steps of every algorithm only improve the loss, this guarantee holds for the final set of returned medoids as well.
>
> To the best of our knowledge, this observation is novel. We have not codified it in our submission and will add this discussion to an appendix.
>
> [G: Submodular Function Maximization, Krause and Golovin]
>
> Additional Experiments: We have also run experiments across two new datasets and metrics to demonstrate the speedups that BanditPAM++ has over BanditPAM. We provide the results for these experiments in our "global" rebuttal in Figure 1.
>
> On both datasets and metrics, we analyze the scaling of BanditPAM++ with respect to both $n$ and $k$. In both additional settings, BanditPAM++ demonstrates significant speedups over BanditPAM. On the scRNA dataset, BanditPAM++ is approximately 6.4x faster than the original BanditPAM algorithm with $k=15$ and $L_1$ distance. On the 20 newsgroups dataset with $k = 15$ and cosine distance (a highly non-Euclidean space), BanditPAM++ is up to 8.5x faster than the original BanditPAM algorithm. Please see Figure 1 and Table 1 of the "global" rebuttal for more details.
>
> We have also run additional experiments to understand the effects of $T$ and $\delta$. Table 2 of our additional results demonstrates that on two different datasets and metrics, $\delta$ can be varied significantly without affecting the quality of the clustering results. Additionally, Figure 2 of our additional experiments shows the loss vs. swap iteration ($T$) for the MNIST dataset with $L_2$ distance and the CIFAR10 dataset with $L_1$ distance, for $k=5, 10$. Increasing $T$ beyond $k$ only leads to marginal decreases in overall loss.

---

> > ### Comment · Reviewer_mktW · 2023-08-14
> >
> > I would like to than the authors for their reply. However, I still have a question regarding their answer for the worst-case.
> >
> > Maybe I am confused, but something about your reply here feels to me a bit too good to be true. First, your submodular function that you maximize is negative, so 0.63 * OPT would actually be better than OPT. I believe the results you cite apply only when the function is nonnegative. Also, this would mean that greedy gives a very good constant factor approximation for the special case of k-means for instance, and as far as I know this is not the case?
> >
> > In summary, could you clarify and point to the precise result for submodular function maximization that you use here?

---

> > > ### Author Response · Authors · 2023-08-14
> > > **Response regarding submodular optimization**
> > >
> > > We apologize for the hasty math and would like to provide a more formal argument below. In addition to negating the loss function (in order to maximize it), we need to add a constant additive factor to make the loss function nonnegative (as Reviewer mktW correctly points out).
> > >
> > > With this new loss function, appropriately negated and with a constant offset added, we can apply the exact result for submodular optimization that says we are 0.63*OPT -- but on the new loss function.
> > >
> > > More formally, we define a new loss function
> > >
> > > $L'(\mathcal{M}) = -L(\mathcal{X}) + max_{x_j \in X} \sum_x d(x_i, xj)$
> > >
> > > $L'(\mathcal{M}) = (-\sum_x \min_{m \in \mathcal{M}} d(x, m)) + max_{x_j \in X} \sum_x d(x_i, xj)$
> > >
> > > Intuitively, this loss if offset by a constant which is equal to the loss induced by the "anti-medoid", i.e., the point which *maximizes* the sum of distances to all other points. Then $L'(\mathcal{M})$ is guaranteed to be nonnegative for any choice of $\mathcal{M}$ and we can apply the greedy optimization result for maximization of nonnegative loss functions to $L'(\mathcal{X})$.
> > >
> > > (While this result may be interesting, we only meant to make this observation in passing; it is not the primary focus of our work.)
> > >
> > > We are unsure of how this implies that greedy gives a very good constant factor for the special case of k-means. Could Reviewer mktW please clarify what they mean here?

---

> > > > ### Comment · Reviewer_mktW · 2023-08-15
> > > >
> > > > Fair enough, but I do not think you would get any interesting guarantee like this. If we denote $D:=\max_{y\in X}\sum_x d(x,y)$, you would get a clustering $C$ such that
> > > >
> > > > $-cost(C)+D\ge 0.63\cdot (-OPT+D)$ which is equivalent to
> > > >
> > > > $cost(C)\le 0.37 D + 0.63\cdot OPT$. If I am not mistaken, $D$ can be arbitrarily worse than the OPT clustering so you do not really get anything using this approach.
> > > >
> > > > For the $k$-means I simply meant that since your problem is a more general formulation (except for the fact that you allows only datapoints,  and I believe you loose only a factor 2 in the objective of $k$-means by taking datapoints only), any result for $k$-medoids would imply a result for $k$-means (allowing only datapoints).
> > > >
> > > > That being said, I agree that this discussion is not the main focus of the paper.

---

> > > > > ### Author Response · Authors · 2023-08-16
> > > > > **Response regarding submodular optimization (2)**
> > > > >
> > > > > Thanks for your comments; we believe you are correct on all counts. In particular, we do not believe that we have made an interesting observation using the submodular line of reasoning.

---

### Official Review · Reviewer_s8tF · 2023-07-02

**Soundness:** 3 good
**Presentation:** 3 good
**Contribution:** 3 good
**Rating:** 6
**Confidence:** 4

**Summary:**

The paper introduces BanditPAM++, a faster version of the BanditPAM algorithm for k-medoids clustering. It incorporates two improvements: the Virtual Arms technique, which reuses loss changes within each SWAP iteration, and the Permutation-Invariant Caching method, which samples reference points and reuses distance computations across steps. The authors provide theoretical guarantees that BanditPAM++ produces the same solution as BanditPAM and showcase its significant speedup on multiple datasets.

**Strengths:**

The originality of the paper lies in the combination of techniques from the multi-armed bandit literature with k-medoids clustering, resulting in a novel algorithm that achieves significant speedup. Most part of the paper is well written and organized.

**Weaknesses:**

1. The presentation of empirical results is somewhat confusing. Here are a few specific concerns:

1.1 The names of methods are not consistent between the text and figures. For example, "BanditPAM" in Line 234 is referred to as "BanditPAM Original with caching" in the figures. It would be helpful to ensure consistency in naming.

1.2 It seems that the results in Figure 1 and Figure 3 are intended to reflect the same experimental setting, but there are inconsistencies. For instance, in the MNIST case with n=70000, the runtime of each method differs between the two plots. Additionally, in the CIFAR10 case, the runtime for k=10 in Figure 3 is significantly smaller than that in Figure 1. It would be beneficial for the author to review and revise the results or provide an explanation for these discrepancies.

1.3 Since the specific speedup of BanditPAM++ can only be evaluated based on the empirical results, it would be clearer to include a table presenting the specific values of speedup. For example, adding a table of average speedup over all dataset sizes on different k. This approach would make it easier for readers to grasp the key points compared to the current figures, which require a non-trivial comparison between several plots.

2. In addition to heuristic methods, there is currently a growing trend in solving clustering problems, including K-Medoids, using global optimization approaches. It would be valuable for the author to discuss these methods in the related work section and provide a comparison between global and heuristic methods. This discussion could highlight the advantages and limitations of each approach, shedding light on the current state-of-the-art in clustering techniques.

**Questions:**

See the weakness

**Limitations:**

The exact speedup of this method cannot be theoretically derived; only empirical speedup can be obtained. Furthermore, the authors have highlighted cases in which the VA technique fails, as mentioned in section 6.

---

> ### Author Rebuttal · Authors · 2023-08-09
>
> We thank Reviewer s8tF for their thoughtful comments and helpful feedback. We provide a point-by-point response below.
>
> Response to W1.1: We apologize for the confusion. The algorithm we refer to as BanditPAM++ is the version of BanditPAM with both the Virtual Arms (VA) technique and the Permutation-Invariant Caching technique, and corresponds to the red lines in all plots. This is our best-performing algorithm. "BanditPAM Original without caching" refers to the original BanditPAM algorithm without the Virtual Arms or Permutation-Invariant Cache techniques, "BanditPAM Original with caching" refers to the original BanditPAM algorithm with a cache but without the Virtual Arms technique, and "BanditPAM VA without caching" refers to the original BanditPAM algorithm with the Virtual Arms technique but without a cache.
>
> We have clarified the legends in the additional experiments presented in the "global" response to reviewers and will clarify the plots in the original submission in our final version.
>
> Response to W1.2: We apologize for the confusion. The reason for this discrepancy is that each set of experiments was run on a different machine, each with different hardware specifications. When run on the same machine, these discrepancies disappear; we will re-run all experiments on a single machine and update the plots for the final version.
>
> Additionally, we have rerun experiments on a single machine for $n=10000$ and $k=5, 10, 15$ on the MNIST dataset with $L_2$ distance, the CIFAR10 dataset with $L_1$ distance, and for two new datasets: the scRNA dataset with $L_1$ distance, and the 20 Newsgroups dataset with cosine distance; our results are presented in Table 1 in the additional experiments in a "global" response to all reviewers. The general trend still persists and we observe that BanditPAM++ is significantly faster than the original BanditPAM algorithm, sometimes even an order of magnitude faster.
>
> On both new datasets and metrics, we analyze the scaling of BanditPAM++ with respect to both $n$ and $k$. In both additional settings, BanditPAM++ demonstrates significant speedups over BanditPAM. On the scRNA dataset, BanditPAM++ is approximately 6.4x faster than the original BanditPAM algorithm with $k=15$ and $L_1$ distance, even with $n = 10000$ on this moderately-sized dataset. On the 20 newsgroups dataset with $k = 15$ and cosine distance (a highly non-Euclidean space), BanditPAM++ is up to 8.5x faster than the original BanditPAM algorithm with $n = 10000$. Please see Figure 1 of the additional experiments in the "global" rebuttal for more details.
>
> We will include a full description of datasets in the final paper.
>
> Response to W1.3: Thank you for this feedback. We have created a Table of the suggested format in Table 1 of our additional experiments and included it in the "global" response to all reviewers. We agree that this is clearer than the plots as currently shown and will include these tables in the final paper. We will also convert the relevant plots in the initial submission to tables when we have enough time to rerun all experiments for different values of $n$.
>
>
> Response to W2: We agree with this feedback and will include global optimization approaches, (e.g., [D] and [E]) in our related work. Our method is crucially different from global optimization approaches such as [D] and [E] in that those methods scale at least quadratically in space and/or time with dataset size. In contrast, our method scales almost linearly in both space and time with dataset size. As such, our heuristic approach scales to even larger datasets, at the expense of a possibly worse optimality gap. We will include this discussion in the final paper.
>
> [D: Global Optimal K-Medoids Clustering of One Million Samples, Jiayang Ren, Kaixun Hua, by Yankai Cao]
> [E: K-Medoids Clustering Is Solvable in Polynomial Time for a 2d Pareto Front]

---

> > ### Author Response · Authors · 2023-08-18
> > **Discussion period**
> >
> > Hi, as the discussion period is drawing to a close, we wanted to ask whether Reviewer s8tF had any additional questions?

---

### Official Review · Reviewer_hYVb · 2023-07-03

**Soundness:** 2 fair
**Presentation:** 2 fair
**Contribution:** 2 fair
**Rating:** 4
**Confidence:** 2

**Summary:**

This paper studies the $k$-mediods clustering problem. Given an instance $P$, the goal of the $k$-mediods problem is to find a subset $C \subseteq P$ of size $k$ such that the sum of the defined distances from data points to $P$ to the centers in $C$ is minimized. In previous heuristic solutions for the $k$-mediods problem, the state-of-the-art $k$-mediods algorithm is the BanditPAM method based on the multi-armed bandit strategy which can return the same solution as PAM (the best heuristic algorithm using exhaustive searching) with high probability and time complexity $O(nlogn)$ and $O(knlogn)$ for each BUILD and SWAP iteration, respectively. In this paper, an improved algorithm is proposed (called BanditPAM++) with faster running time and can return the same solution to BanditPAM with high probability. By designing Virtual Arms (VA) technique and using a permutation-invariant cache (PIC) structure, this paper shows that the total number of distance computations in expectation can be reduced to $O(nlogn)$, which is much faster than BanditPAM method. Experimental results show that the proposed BanditPAM++ method runs 10 times faster than BanditPAM method with the same clustering results on some of the commonly used datasets for clustering tasks.

**Strengths:**

1. This paper proposes a new heuristic method for the $k$-mediods clustering problem. The authors show that  the proposed method runs much faster than the current SOTA (BanditPAM method) and can give the same solution as the current SOTA with high probability.

2. This paper proposed two novel techniques and structures, namely the Virtual Arms (VA) technique and the permutation-invariant cache (PIC), which is proved to be efficient for accelerating the BanditPAM method in both theory and practice.

3. The experimental results show that the proposed BanditPAM++ method has significant advantages on the running time compared with BanditPAM method, while the clustering quality of BanditPAM++ matches that of BanditPAM

**Weaknesses:**

1. The theoretical proofs of this paper is mainly based on the assumptions that (1): during the SWAP process, the samples are drawn from a sub-Gaussian distribution. (2) the maximum number of swap steps is limited. However, there is no detailed discussion about the assumptions made (especially the sub-Gaussian distribution assumption). Although the authors mentioned in the conclusions and limitations part that "while the assumptions of BanditPAM and BanditPAM++ are likely to hold in many practical scenarios, it is important to acknowledge that these assumptions can impose limitations on our approach", there is no clear explanation about the assumptions made and it is unclear why the Gaussian distribution is likely to hold in many practical scenarios. Besides, in a clustering algorithm, the iteration rounds needed to reach the convergence is not usually a constant $T$. For example, in local search algorithms for clustering, the iteration rounds could be as large as $O(klog\Delta)$ if the initialization is bad, where $\Delta$ is the aspect ratio of the given instance. Thus, it is unlcear how the number of swap steps could affect the results of BanditPAM++

2. For the proposed BanditPAM++ method, this paper claims that it runs much faster than BanditPAM. However, there is no direct comparison between each phase (i.e., the BUILD and SWAP iteration). It is unclear how the time complexity is improved compared with BanditPAM method where each BUILD iteration requires $O(nlogn)$ running time and each SWAP iteration requires $O(nklogn)$ running time.

**Questions:**

1. Can the authors provide detailed comparison between the running time of BanditPAM++ and BandiPAM for both BUILD iteration and SWAP iteration.

2. Does the maximum iteration $T$ required for BanditPAM and BanditPAM++ to reach a good convergence the same?

3. Why the authors claim that the assumptions of BanditPAM and BanditPAM++ are likely to hold in many practical scenarios?

4. If the sub-gaussian distribution assumptions do not hold, what will the guarantee be like (maybe some worse guarantee)?

**Limitations:**

I don't think this paper has potential negative societal impact of their work.

---

> ### Author Rebuttal · Authors · 2023-08-09
>
> We thank Reviewer hYVb for their thoughtful comments and helpful feedback. We provide a point-by-point response below.
>
> Response to W1, Q3, and Q4 (sub-Gaussianity): We agree that it is ideal to remove the sub-Gaussian assumption and discuss this limitation in the revised paper. However, we argue this is a reasonable assumption for many real-world datasets. First, this assumption holds in any bounded dataset, since a distribution with bounded support is always $\sigma$-sub-Gaussian for some $\sigma$. Furthermore, prior work [20, Appendix 1.3] has shown that these assumptions hold in most practical datasets, e.g., the MNIST dataset with either $L_1$ or $L_2$ distances and the scRNA dataset with $L_1$ distance. Similar observations have been made in [3, Figures 2 and 3]
>
> Our assumptions also follow that of the original BanditPAM paper [20], so are as general as in the original BanditPAM algorithm. These assumptions are also common in the multi-armed bandit literature [2, 3]. The scaling of our algorithms in Section 5 also demonstrates that the sub-Gaussianity assumptions hold in practice. We acknowledge, however, that these assumptions may introduce a limitation in pathological datasets, e.g., datasets that are unbounded.
>
> If the sub-Gaussianity assumptions do not hold, it is possible that, in the worst case, BanditPAM++ will not identify the correct medoids. However, this is the same behavior as BanditPAM and in practice, we do not observe that this happens. Our results in Section 5 and prior work [20] demonstrate that BanditPAM++ and BanditPAM return the same results as PAM.
>
> Response to W1 and Q2 ($T$): We have run additional experiments to understand the effect of $T$ and provided them in the "global" response to all reviewers.
>
> Note that increasing $T$ will decrease the overall loss of the clustering solution as more swaps can be performed; however, we find that beyond $T = k$, increasing $T$ further does not lower the overall loss very significantly. This is described on Line 121 of our original submission and has been observed in prior work [19, 20].
>
> We have provided new empirical evidence for this claim in the additional experiments in the "global" rebuttal. Figure 2 of our additional experiments shows the loss vs. swap iteration ($T$) for the MNIST dataset with $L_2$ distance and the CIFAR10 dataset with $L_1$ distance, for $k=5, 10$. Increasing $T$ beyond $k$ only leads to marginal decreases in overall loss. BanditPAM++ and BanditPAM track the same optimization trajectory, so are superimposed.
>
> Finally, we note that setting a maximum number of iterations $T$ for iterative algorithms is common practice, e.g., in $k$-means. We are aware of convergence results for $k$-means, (e.g., [F]), however it is not immediately clear how these carry over to $k$-medoids. We acknowledge that we do not provide a strict theoretical bound on $T$ for the convergence of $k$-medoids algorithm, however, to the best of our knowledge no such bound exists for PAM or BanditPAM either (which would imply a bound for BanditPAM++, since they all follow the same optimization trajectory). Proving such a bound on $T$ has been observed to be a difficult problem [19, 20].
>
> We also note that the number of iterations required for BanditPAM and BanditPAM++ to reach a good solution are the same; this is because BanditPAM++ is guaranteed to return the same solution as BanditPAM to every SWAP step. This is the key point of Theorem 3, which states that BanditPAM++ follows the same optimization trajectory as BanditPAM and PAM. However, we agree that this is currently unclear as written because Theorem 4 states that only the final set of $k$ medoids is the same. We will clarify that BanditPAM++ tracks the exact same optimization trajectory as BanditPAM and PAM -- in particular, for each BUILD and SWAP step -- in the final paper.
>
> [F: "A theoretical analysis of Lloyd's algorithm for k-means clustering" Bhowmick 2009]
>
> Response to W2 and Q1: Thank you for the comment. We have run additional experiments to understand the speedups in the BUILD step only, the SWAP steps only, and in the BUILD + SWAP steps. We present our results in Table 1 in the additional experiments in a "global" response to all reviewers.
>
> We observe that the gains for the BUILD step are slightly less than those for the SWAP steps; this is expected because the Virtual Arms (VA) technique improves only the SWAP step and not the BUILD step. However, we still observe significant gains in the BUILD step due to the Permutation-Invariant Caching (PIC) technique, which caches distance computations and reuses them across iterations of the BUILD step. For example, on the scRNA dataset with $L_1$ distance, the BUILD step of BanditPAM++ is still over $3\times$ faster than BanditPAM with $k=15$. On the 20 Newsgroups dataset with cosine distance, the BUILD step of BanditPAM++ is still $3\times$ faster than the BUILD step of BanditPAM when $k=15$.
>
> In the same table, we also provide results analyzing the improvements to the SWAP step alone, as well as the overall algorithm (BUILD + SWAP).
>
> On both new datasets and metrics, we analyze the scaling of BanditPAM++ with respect to both $n$ and $k$. In both additional settings, BanditPAM++ demonstrates significant speedups over BanditPAM. On the scRNA dataset, BanditPAM++ is approximately 6.4x faster than the original BanditPAM algorithm with $k=15$ and $L_1$ distance, even with $n = 10,000$ on this moderately-sized dataset. On the 20 newsgroups dataset with $k = 15$ and cosine distance (a highly non-Euclidean space), BanditPAM++ is up to 8.5x faster than the original BanditPAM algorithm with $n = 10,000$.
>
> We will include a full description of datasets in the final paper.

---

> > ### Author Response · Authors · 2023-08-18
> > **Discussion period**
> >
> > Hi, as the discussion period is drawing to a close, we wanted to ask whether Reviewer hYVb had any additional questions?

---

### Official Review · Reviewer_zP71 · 2023-07-05

**Soundness:** 3 good
**Presentation:** 2 fair
**Contribution:** 2 fair
**Rating:** 4
**Confidence:** 4

**Summary:**

The authors developed an algorithm (called BanditPAM++) which is claimed to be more efficient than banditPAM, the state-of-the-art algorithm for k-medioid (which is the version of the k-means problem where the centers must be input points). They prove that with high probability the two algorithms produce the same results, with BanditPAM++ being more efficient. An experimental evaluation on two datasets complements their results.

**Strengths:**

- the work focuses on improving the scalability of an effective algorithm for clustering
- theoretical results support their claims, which are validated empirically on two datasets

**Weaknesses:**

- it is not clear how some of the parameters (in particular T=max number of swaps and delta) affect the results.
- only two datasets are included in the experimental evaluation. I recommend a more extensive evaluation, given that it is not clear what is the advantage of their approach from a theoretical point of view
- the analysis focuses on the case when the input points follow a subgaussian distribution, while for example k-means++ guarantees hold for any distribution
- the presentation of the paper could be improved (e.g. the role of delta is not discussed, not immediately clear which curve in the experiments correspond to BanditPAM++, some claims are inaccurate, e.g. there is a version of k-means that can deal with cosine similarity)


**Questions:**

Questions:
- from a theoretical point of view, what is the advantage of BanditPAM++ over BanditPAM? (both have running time O(n logn) as far I could check). Update after rebuttal: I had missed that BanditPAM++ is faster than BanditPAM by a factor of k. This should be stressed in the text. Perhaps a short paragraph right after Theorem 4 might help.
- discuss the role of delta and the max number of swaps T and how they might affect the results

other comments:
- Figure 4 does not offer many insights, all approaches are superimposed and we just see a straight line.
- there is a version of k-means with cosine similarity, in contrast with what is claimed in the paper (references [4,6,18] are old and not up to date)

A Faster Sampling Algorithm for Spherical k-means, Journal of machine learning 2018.

- I recommend to evaluate the impact of delta and T in the experiments, for future work
- in the experiments it should be immediately clear which curves correspond to BanditPAM++ (all of the variants? some?)

**Limitations:**

No potential negative societal impact, as far as I could check.

---

> ### Author Rebuttal · Authors · 2023-08-09
>
> We thank Reviewer zP71 for their thoughtful comments and helpful feedback. We provide a point-by-point response below.
>
> Response to W1 and Q1: From a theoretical point of view, the computational complexity of BanditPAM++ is same as that of BanditPAM with respect to the number of datapoints $n$ (both $O(n$ log $n)$), but it is smaller than that of BanditPAM with respect to the number of medoids $k$,  ($O(1)$ vs. $O(k)$) in each SWAP. Overall, BanditPAM++ is $O(k)$ faster than BanditPAM in each SWAP step.
>
> For real-data, wall-clock time of BanditPAM++ may slightly increase with $k$, possibly because the MAB problem instances became harder (smaller $\Delta_i$'s) as $k$ increases (assumed not to change in theoretical analysis). Nonetheless, this will have the same impact on both algorithms. Therefore, the improvement of BanditPAM++ over BanditPAM is still $O(k)$.
>
> In addition, on real-data experiments, we verified that BanditPAM++ is substantially faster than BanditPAM while producing same results and that this improvement increased with $k$ (e.g., $4.75\times$ faster with $k=5$, $8.23\times$ faster with $k=10$, and $10.77\times$ faster with $k=15$ on the same dataset across various dataset sizes; Table 1 of the additional rebuttal experiments. Figure 3b of the original submission and Figure 1 of the additional rebuttal experiments show similar trends).
>
> Response to W2 and Q2: We have run additional experiments to understand the effects of $T$ and $\delta$ and provided them in the "global" response to all reviewers.
>
> We note that the choice of hyperparameter $\delta = n^{-3}$ is not very important and is only chosen to show that the correctness probability is $1 - o(1)$; an analogous claim can be made for arbitrary $\delta$ (Remark A1 on Lines 407-40). In practice, even setting $\delta$ very large does not affect the quality of our results. We present an additional experiment demonstrating this in the "global" rebuttal. Table 2 of our additional results demonstrates that on two different datasets and metrics, $\delta$ can be varied significantly without affecting the quality of the clustering results.
>
> Note that increasing $T$ will decrease the overall loss of the clustering solution as more swaps can be performed; however, we find that beyond $T = k$, increasing $T$ further does not lower the overall loss very significantly. This is specified on Line 121 of our original submission and has been observed in prior work [19, 20]. Figure 2 of our additional experiments shows the loss vs. swap iteration ($T$) for the MNIST dataset with $L_2$ distance and the CIFAR10 dataset with $L_1$ distance, for $k=5, 10$. Increasing $T$ beyond $k$ only leads to marginal decreases in overall loss.
>
> Response to W3: We emphasize that from a theoretical point of view, BanditPAM++ is $O(k)$ faster than BanditPAM in each SWAP.
>
> We have also run new experiments across two different datasets and metrics ($L_1$ distance on scRNA-seq data; cosine similarity on newsgroups data) to demonstrate the speedups that BanditPAM++ has over BanditPAM in non-Euclidean spaces. On both new datasets and metrics, BanditPAM++ demonstrates significant speedups over BanditPAM. Please see Figure 1 and Table 1 in the additional experiments.
>
> Response to W4: We followed the same assumptions of the original BanditPAM paper [20]. The subgaussian assumption is a reasonable assumption in many real-world datasets; for example, these assumptions hold in any bounded dataset. Furthermore, prior work [20, Appendix 1.3] has shown that these assumptions hold in most practical datasets, e.g., the MNIST dataset with either $L_1$ or $L_2$ distances and the scRNA dataset with $L_1$ distance. Similar observations have been made in [3, Figures 2 and 3]
>
> These assumptions are also common in the multi-armed bandit literature [2, 3]. The scaling of our algorithms in Section 5 also demonstrates that the sub-Gaussianity assumptions hold in practice.
>
> However, we also acknowledge that more general assumptions may be more desirable, e.g., for unbounded or heavy-tailed distributions. We added a discussion of this limitation in the revised paper.
>
> Response to W5 (curves): The algorithm we refer to as BanditPAM++ is the version of BanditPAM with both the Virtual Arms (VA) technique and the Permutation-Invariant Caching (PIC) technique, and corresponds to the red lines in all plots. This is our best-performing algorithm. "BanditPAM Original without caching" refers to the original BanditPAM algorithm without the VA or PIC techniques, "BanditPAM Original with caching" refers to the original BanditPAM algorithm with a cache but without the VA technique, and "BanditPAM VA without caching" refers to the original BanditPAM algorithm with the VA technique but without PIC.
>
> We have clarified the legends in the additional experiments presented in the "global" rebuttal and will clarify the plots in the original submission in our final version.
>
> Response to W5 and Q4 (k-means comments): Thank you for pointing this out. We will clarify this and cite "A Faster Sampling Algorithm for Spherical k-means, Journal of machine learning 2018." in our final paper.
>
> We also would like to note that the general point is still valid: $k$-means typically requires specific distance metrics for efficient solutions. In contrast, $k$-medoids is agnostic to the distance metric being used. In fact, the distance function being used in $k$-medoids need not even be a metric and can be an arbitrary dissimilarity function (i.e., may violate the triangle inequality, may be negative, and may not be symmetric). For this reason, $k$-medoids has been used to cluster "exotic" objects such as trees [20].
>
> Response to Q3: Thank you for this feedback. We will convert Figure 4 to two much smaller tables that show the losses of each algorithm relative to BanditPAM, where the entries of each table will simply be "1.00x". This is also what we have done for Table 2 of our additional rebuttal experiments.

---

> > ### Author Response · Authors · 2023-08-18
> > **Discussion period**
> >
> > Hi, as the discussion period is drawing to a close, we wanted to ask whether Reviewer zP71 had any additional questions?

---

> > > ### Comment · Reviewer_zP71 · 2023-08-20
> > > **Discussion Period**
> > >
> > > I would like to thank the author for their clarifications.

---

> > > > ### Author Response · Authors · 2023-08-21
> > > > **Response to Reviewer zP71**
> > > >
> > > > Thank you for taking the time to review our paper and rebuttal.
> > > >
> > > > We received an email containing the original comment text of Reviewer zP71's original comment, which emphasized that they felt the paper was not yet ready for publication. If Reviewer zP71 is able to provide any additional feedback, we would very happily receive it, especially in case we need to resubmit the paper to a later conference.

---

### Official Review · Reviewer_oPsi · 2023-07-06

**Soundness:** 3 good
**Presentation:** 2 fair
**Contribution:** 3 good
**Rating:** 7
**Confidence:** 2

**Summary:**

The paper
 * presents a new algorithm for the k-medoids problem called BandidPAM++, which is an extension of BanditPAM
 * analyzes the algorithm theoretically showing that it produces essentially the same output as BanditPAM and uses O(n log n) distance computations
 * evaluates the algorithm empirically comparing it to BanditPAM

**Strengths:**

* The paper new algorithm outperforms the baseline in experiments, delivering significantly better running times
* While I'm not absolutely certain, it appears to me that some of the ideas involved are nontrivial

**Weaknesses:**

The main issue with the paper is the presentation. In the current form the description of the proposed algorithm consists of a short, high-level description + a nontrivial pseudocode. This is especially disappointing given that it would be easy to fit in more text in the first 9 pages (there are many subsection headers, figures could be made smaller, Algorithm 1 could potentially go to the appendix, etc). This, combined with the fact that I'm not familiar with the area of bandit algorithms, makes it difficult for me to make a confident decision.

My second (smaller) concern is about the empirical evaluation: while the motivation for using k-medoids is clustering points in non-Euclidean spaces, the experiments use two datasets, both of which define Euclidean space, which is non-ideal (but I guess acceptable for a paper whose main goal is to demonstrate speedups)

[Edit: thank you for addressing this point in the rebuttal]

**Questions:**

1. Is it true that BanditPAM and BanditPAM++ come with the same running time bound on the BUILD iterations?
1. Is the running time of each SWAP iteration O(k n log n) for BanditPAM vs O(n log n) for BanditPAM++? (Theorem 4 bounds the number of comparisons which is not the same as the running time)
1. Do you use any results from the multi-arm bandit literature in any of the proofs?
1. Since Theorem 1 is a reformulation of a known fact, what is the novelty in the virtual arms optimization?
1. How similar is Theorem 3 to what was previously done for BanditPAM?
1. What is the high-level idea behind Theorem 3?
1. Could you also elaborate on the main idea behind permutation-invariant cache? I find the current description (lines 196-201) to be too compact to be understandable.
1. Line 131 suggests that each $D_j$ is a random variable which represents a *uniform* distribution over $\mathcal{S}$. Could you formally define $D_j$?
1. Does Table 1 describe BanditPAM or BanditPAM++? The caption vs line 183 seem to contradict each other.

**Limitations:**

Yes

---

> ### Author Rebuttal · Authors · 2023-08-09
>
> We thank Reviewer oPsi for their thoughtful comments and provide a point-by-point response below.
>
> Weakness 1: We have now added detailed explanations of the pseudocode at the end of Section 3 and made better use of space. BanditPAM++ improves upon BanditPAM via two techniques: the Virtual Arms (VA) technique, and Permutation-Invariant Caching (PIC). The Virtual Arms technique improves the sample complexity of BanditPAM++ relative to BanditPAM by $O(k)$ in each SWAP. The Permutation-Invariant Caching technique results in improvements to wall-clock time.
>
> The Virtual Arms technique is implemented in lines 6-11 of Algorithm 2. Specifically, for a given reference point $x_r$ and a candidate point $x_c$, instead of evaluating all $k$ medoids as in BanditPAM (Line 6 of Algorithm 1), BanditPAM++ only performs 3 distance computations (Line 7 of Algorithm 2), which suffices to update the information of all $k$ medoids. This brings an $O(k)$ improvement of algorithm complexity and is permitted by Theorem 1.
>
> Permutation-Invariant Caching is implemented in line 5 of Algorithm 2. Instead of randomly sampling a reference point $x_r$ as in BanditPAM (line 5 Algorithm 1), BanditPAM++ prestores a permutation of reference points in cache and draws the reference point $x_r$ based on this prestored permutation. This procedure substantially increases the speed of BanditPAM++ in wall-clock time. Choosing a permutation of datapoints a priori and using this permutation across all phases of BanditPAM++ increases the proportion of cache hits when the cache is too small to hold all $n^2$ pairwise distances. Without a permutation order, many cache misses would occur because we are not reusing the same reference points (and therefore distances) across different steps of the algorithm.
>
> Weakness 2: The CIFAR experiment in the original submission used $L_1$ distance (lines 241-242 and captions of Figures 1, 3, and 4 of the original submission).
>
> We have also run new experiments across two different datasets and metrics ($L_1$ distance on scRNA-seq data; cosine similarity on newsgroups data) to demonstrate the speedups that BanditPAM++ has over BanditPAM in non-Euclidean spaces. BanditPAM++ demonstrates significant speedups over BanditPAM (Figure 1 and Table 1 in the additional experiments).
>
> Q1: BanditPAM and BanditPAM++ have the same sample complexity for BUILD, because both use Algorithm 3 (in Appendix 1 of our original submission) for the BUILD phase. However, BanditPAM++ accelerates its BUILD steps using the Permutation-Invariant Caching technique described above to cache the distance computations and reuse them across iterations, so its wall-clock time is better (Table 1 of the additional rebuttal experiments).
>
> Q2: Yes, the running time of each SWAP iteration is $O(k n \text{log} n)$ for BanditPAM vs $O(n \text{log} n)$ for BanditPAM++.
>
> We chose to focus on the number of distance computations in our theoretical analysis because the overall complexity is governed by the number of distance computations [19, 20]. We also evaluated the wall-clock time improvement using real-data experiments. As shown in Figure 3, the improvement of BanditPAM++ over BanditPAM increases with $k$.
>
> For real-data, wall-clock time of BanditPAM++ also slightly increased with $k$, possibly because the MAB problem instances became harder (smaller $\Delta_i$'s) as $k$ increased (assumed to be the same in theoretical analysis). Nonetheless, this will have the same impact on both algorithms. Therefore, the improvement of BanditPAM++ over BanditPAM is still $O(k)$.
>
> Q3: Yes, we borrowed some proof techniques from the multi-armed bandit literature. Algorithms 1 and 2 (as well as Algorithm 3 in Appendix 1) are similar to the successive elimination algorithm from multi-armed bandits. Our proof of Theorem 3, which proves correctness and the sample complexity of BanditPAM++, is similar to the proof of correctness and sample complexity for successive elimination. Additionally, the proof of Theorem 4 relies on proofs from the multi-armed bandit literature to convert a gap-dependent bound that depends on the $\Delta_i$'s to a bound in terms of the dataset size $n$ [2, Appendix 2 of 3].
>
> Q4: Though Theorem 1 has been described elsewhere [19], our paper provided its first formal proof.  More importantly, it is non-trivial to incorporate the observation in Theorem 1 into a randomized algorithm, BanditPAM, to achieve a complexity improvement. Before our work, it was unclear that the observation in Theorem 1 could be applied to a randomized algorithm for algorithmic speedups.
>
> Q5: The proofs are different because BanditPAM considers datapoint-medoid pairs as separate “arms” whereas BanditPAM++ considers the $k$ medoids of a given datapoint share the same “arm” and are updated together. As a result, the proof from BanditPAM does not carry over to BanditPAM++.
>
> Q6: The  idea behind Theorem 3 is that BanditPAM++ is a) correct, and b) sample-efficient. Regarding a), Theorem 3 shows that with high probability, BanditPAM++ returns the same results as BanditPAM (and therefore PAM). a) is achieved by a union bound over the error probability of all confidence interval estimates. Regarding b), the bound is a sum of sample complexity over all datapoints (arms). Specifically, for a datapoint $x$, Theorem 3 states that its sample complexity depends on the difference between its quality and that of the best arm, i.e., the gap $\Delta_x$'s. "Bad" arms with high $\Delta_x$ will be thrown out with just a few samples, whereas "good" arms with low $\Delta_x$ will be sampled more to determine the true best arm.
>
> Q7: Please see response to Weakness 1.
>
> Q8: We define $D_j$ as an r.v. which has a uniform distribution over the possible distances from point $j$ to any point in the dataset; in this way, $E[D_j] = \mu_j$ is the average distance from point $j$ to all points in the dataset.
>
> Q9: Line 183 should read "BanditPAM" instead of "BanditPAM++"; we have fixed this.

---

> > ### Comment · Reviewer_oPsi · 2023-08-14
> > **Thank you for the response**
> >
> > Just one follow up question: in the response you mention that you have updated the write-up. Was the updated version included in the response? (I do not see it anywhere)

---

> > > ### Author Response · Authors · 2023-08-14
> > > **Updated writeup**
> > >
> > > Hi, we did not initially include an updated version in the response as the instructions on https://neurips.cc/Conferences/2023/CallForPapers prohibit this:
> > >
> > > > *Author responses*: ... Authors may not submit revisions of their paper or supplemental material, but may post their responses as a discussion in OpenReview.
> > >
> > > However, we would be happy to provide the updated version of our paper if this would not break the rules. Could Reviewer oPsi (and/or the AC) confirm that providing a revision would NOT break the rules? If so, we are happy to upload the revision as an anonymized PDF link.
> > >
> > > If providing a revision of the paper is NOT allowed, we would also be happy to copy and paste the relevant sections into "Official Comments" here.

---

> > > > ### Author Response · Authors · 2023-08-18
> > > > **Discussion Period**
> > > >
> > > > Hi, as the discussion period is drawing to a close, we wanted to ask whether Reviewer oPsi had any additional questions? Additionally, would it be permissible to post a link to an updated version of the paper?

---

### Author Rebuttal · Authors · 2023-08-09

Attached please find additional experimental results that we refer to in our rebuttals to reviewers.

---

### Decision · Program_Chairs · 2023-09-21

**Decision:**

Accept (poster)

**Comment:**

The paper presents a new algorithm for the k-medoids clustering. The algorithm improves on previous work in the area, in particular on the  state-of-the-art algorithm, the BanditPAM algorithm.

The ideas introduced in the paper are interesting and the experimental results nice but only on Euclidean datasets. This is a bit surprising because the k-medoids problem is interesting mainly in metric spaces. Nevertheless the theoretical results and the improvement on the running time are interesting and worth publishing.

For the final version we would suggest to extend the paper with:
- an in-depth discussion on the fact that the results hold on under some assumptions
- some experiment on non-euclidean datasets